# A robust model for cell type-specific interindividual variation in single-cell RNA sequencing data

Minhui Chen [1] ✉ & Andy Dahl [1] ✉

Single-cell RNA sequencing (scRNA-seq) has been widely used to characterize cell types based on their average gene expression profiles. However, most studies do not consider cell type-specific variation across donors. Modelling this cell type-specific inter-individual variation could help elucidate cell type-specific biology and inform genes and cell types underlying complex traits. We therefore develop a new model to detect and quantify cell type-specific variation across individuals called CTMM (Cell Type-specific linear Mixed Model). We use extensive simulations to show that CTMM is powerful and unbiased in realistic settings. We also derive calibrated tests for cell type-specific inter-individual variation, which is challenging given the modest sample sizes in scRNA-seq. We apply CTMM to scRNA-seq data from human induced pluripotent stem cells to characterize the transcriptomic variation across donors as cells differentiate into endoderm. We find that almost 100% of transcriptome-wide variability between donors is differentiation stage-specific. CTMM also identifies individual genes with statistically significant stage-specific variability across samples, including 85 genes that do not have significant stage-specific mean expression. Finally, we extend CTMM to partition interindividual covariance between stages, which recapitulates the overall differentiation trajectory. Overall, CTMM is a powerful tool to illuminate cell type-specific biology in scRNA-seq.

The technology of single-cell RNA sequencing (scRNA-seq) profiles gene expression at the resolution of single cells. This resolution may be essential for understanding molecular mechanisms underlying many complex traits because disease gene expression is highly cell type-specific[1–3]. For example, *APOE* is a risk gene for Alzheimer's disease that is downregulated in astrocytes but is upregulated in microglia[2]. One common application of scRNA-seq is to investigate differentially expressed genes (DEG) that exhibit differences in mean expression between cell types, such as diseased vs healthy[4] or pre- vs post-treatment[5–7]. Furthermore, methods to infer cell type labels in scRNA-seq data primarily rely on differential mean expression between cell types[8,9].

Several studies have applied linear mixed models to scRNA-seq data to account for variance across individuals, cell types, or experimental batches[10–14], but this variance has not been unbiasedly partitioned across cell types, and the potential for bias and miscalibration have not been evaluated. Understanding this variation could help identify and characterize genes and cell types that cause interindividual variation in complex traits ranging from height to autoimmune disorders. Studies using bulk RNA-seq have shown that gene expression variability informs disease biology and drug development[15–17]. However, bulk transcriptomics has poor resolution on individual-cell types, which can cause both false positives and false negatives. In particular, prior signals in bulk RNA-seq could be

[1]Section of Genetic Medicine, University of Chicago, Chicago, IL 60637, USA. ✉e-mail: minhuic@uchicago.edu; andywdahl@uchicago.edu

explained by variation in cell type proportions rather than variation in gene expression within cell types[18]. Because scRNA-seq data has a cell-level resolution, it provides an opportunity to powerfully partition expression variation within and between cell types. This has recently become possible with the proliferation of population-scale scRNA-seq studies that contain hundreds of individuals[13,19–22].

In this paper, we develop CTMM (Cell Type-specific linear Mixed Model) to detect and quantify cell type-specific variation across individuals in scRNA-seq data. We performed a series of simulations to evaluate CTMM's performance in a broad range of realistic settings. We then applied CTMM to characterize transcriptomic variation across individual donors along the developmental trajectory from human induced pluripotent stem cells (iPSCs) to endoderm. Transcriptome-wide, CTMM found that almost all interindividual variation was specific to each developmental time point, and the Full model found greater correlations between nearby time points. We also identified specific genes with statistically significant time point-specific variation across individuals, including genes with known importance for cell pluripotency and differentiation. Finally, we studied the recent data from the OneK1K cohort and found that CTMM can be applied to this kind of large-scale, low-depth sequencing data.

## Results

### Overview of CTMM

CTMM is a linear mixed model that partitions single-cell gene expression variation across individuals into two distinct components: variation shared across cell types and variation specific to each cell type. We fit CTMM to cell type-specific pseudobulk (CTP) data, which is the mean expression over cells within each cell type for each individual. For a given gene, the CTP expression for individual $i$ and cell type $c$ is:

$$\mathbf{y}_{ic} := \frac{1}{n_{ic}} \sum_{s=1}^{n_{ic}} y_{ics} \qquad (1)$$

where $y_{ics}$ is the gene expression level for the $s$-th cell from cell type $c$ in individual $i$ and $n_{ic}$ is the number of cells in individual $i$ from cell type $c$. CTMM models the CTP expression data by:

$$\mathbf{y}_{ic} = \boldsymbol{\beta}_c + \boldsymbol{\alpha}_i + \boldsymbol{\Gamma}_{ic} + \boldsymbol{\delta}_{ic} \qquad (2)$$

Here, $\boldsymbol{\beta}_c$ is the mean expression level in cell type $c$, which we model as a fixed effect. $\boldsymbol{\alpha}_i$ captures differences between individuals that are shared across cell types, which we model as a random effect: $\boldsymbol{\alpha}_i \sim^{iid} N(0, \sigma_\alpha^2)$. $\boldsymbol{\Gamma}_{ic}$ captures the difference between individuals that is specific to cell type $c$, which we also model as a random effect: $\boldsymbol{\Gamma}_i \sim^{iid} N(0, \mathbf{V})$, where $\boldsymbol{\Gamma}_i$ is a vector of cell type-specific expression for individual $i$ across all $C$ cell types and $\mathbf{V}$ is a $C \times C$ matrix describing cell type-specific variances and covariances across cell types. $\boldsymbol{\delta}_{ic}$ is the noise due to measurement errors at single cells and/or variation from cell subtypes, which we model by: $\boldsymbol{\delta}_{ic} \sim^{ind} N\left(0, \frac{\sigma_{ic}^2}{n_{ic}}\right)$, where $\sigma_{ic}^2$ is cell-to-cell variance within individual $i$ and cell type $c$. We estimate this quantity from the cell-level data by $\sigma_{ic}^2 := \sum_{s=1}^{n_{ic}}(y_{ics} - y_{ic})^2/(n_{ic}-1)$. In the Methods, we show how this model is derived from a single cell-level model, which also motivates our Gaussian assumption on $\boldsymbol{\delta}_{ic}$ based on the central limit theorem. We also developed a version of CTMM that applies to overall pseudobulk (OP) data, which averages over all cells from all cell types for each individual. This is a useful analogy to bulk sequencing data, but we find that it is far less powerful in our setting.

The focus of CTMM is on the covariance matrix $\mathbf{V}$, which captures cell type-specific variation across individuals. We consider three nested models of cell type-shared and -specific variation defined by the structure of $\mathbf{V}$. In the simplest model where $\mathbf{V} = 0$, all variation is shared homogeneously between cell types ("Hom"), with cell types differing only in mean expression. The next model allows independent variation

in each cell type ("Free"), i.e., cell type-specific variation, by allowing $\mathbf{V}$ to be an arbitrary diagonal matrix. The richest model allows for arbitrary forms of covariance between cell types ("Full"), where $\mathbf{V}$ can be any arbitrary semidefinite matrix and $\sigma_\alpha^2 = 0$ for identifiability (Methods).

We explored several statistical methods to fit and test CTMM's parameters. Achieving calibrated and unbiased estimates in CTMM is challenging because scRNA-seq datasets currently have small to moderate numbers of donors, ranging from one to hundreds, and thus off-the-shelf asymptotic tests may fail. We implemented three approaches to fit CTMM: maximum likelihood (ML), restricted maximum likelihood (REML), and method-of-moments (HE, as it is called Haseman-Elston regression in genetics). Then, we implemented the likelihood ratio test (LRT) and Wald tests to compare the Free and Hom models, which tests whether interindividual variation is cell type-specific or shared uniformly across cell types. Importantly, we develop a novel testing framework based on jackknife (JK) to address false positives in established tests that arise from the complexity of scRNA-seq data (Methods).

### Simulation

We simulated a series of scenarios to assess the performance of CTMM. We simulated Hom and Free models by varying sample size, level of cell type-specific variance, number of cell types, and cell type proportions. We first evaluated the accuracy to quantify cell type-specific variance. Supplementary Fig. 1 showed the estimation of cell type-specific variance in the simulation of the Free model with varying sample sizes from 20 to 1000. As expected, when fitting simulated data into the Free model, both OP and CTP performed well, as illustrated by the roughly unbiased estimates of cell type-specific variance $\mathbf{V}$. The performance improved along with the increase in sample size. CTP provided more precise estimates than OP, since CTP uses more information than OP by modeling pseudobulk expression for each cell type. Comparing methods for parameter estimation, likelihood-based methods, including ML and REML had similar level of precision, and both had better precision than HE, since likelihood-based methods utilize more information than HE by assuming normal distribution of random effects. Supplementary Fig. 2 showed estimates with varying levels of cell type-specific variance. Our models provided unbiased estimates of cell type-specific variance, even in the simulation of the Hom model where there is no cell type-specific effect. Additionally, Supplementary Fig. 3 showed that CTMM is unbiased across different numbers of cell types. Supplementary Fig. 4 showed estimates with varying cell type proportions. When decreasing the proportion of the main cell type (with the largest cell type-specific variance), all models performed well except for HE with CTP input, which broke down when the main cell type proportion went below 10%. We also simulated under the Full model, which had precise and unbiased estimates of covariance between cell types when the sample size was above 50 with CTP (Supplementary Fig. 5).

We then evaluated the power of our models to detect cell type-specific variance. Figure 1 showed positive rates of REML and HE using OP or CTP data as input for different sample sizes. Under the simulation of the Hom model where there is no cell type-specific variance, different tests for cell type-specific variance with both OP and CTP input were appropriately null with around 5% of the false positive rate, except for REML (Wald), that is Wald test in REML using precision matrix inferred from the Fisher information matrix. REML (JK), that is jackknife-based Wald test in REML, and HE were inflated in CTP when sample size was 50 or lower. Under the simulation of Free model, CTP gained much larger power than OP, for example, when sample size was 50, CTP had tenfold positive rate over OP (100 versus 10% using REML with LRT). REML (LRT) in CTP had the best power. Its true positive rate reached above 80% even when the sample size was only 20 and reached 100% when the sample size was 50. The other three tests in

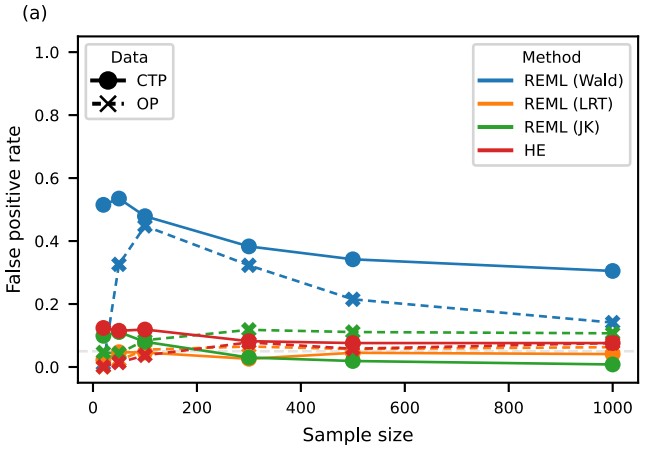

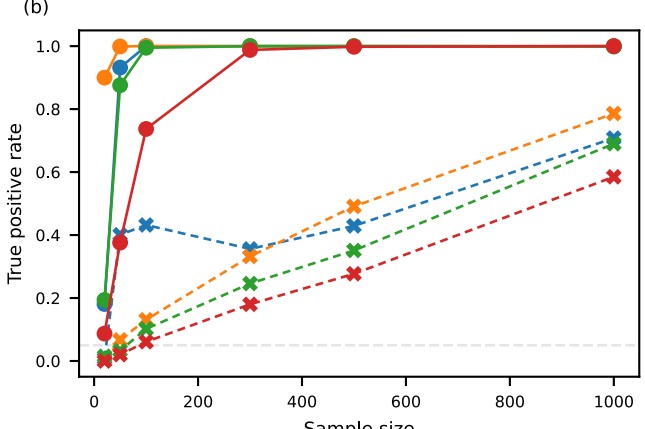

**Fig. 1 | Power of CTMM's test of cell type-specific variance in simulations with varying sample sizes. a** In the simulation of the Hom model, there is no cell type-specific variance; **b** In the simulation of the Free model, each cell type has its own

cell type-specific variance. CTP cell type-specific pseudobulk, OP overall pseudo-bulk, REML (Wald) REML with Wald test, REML (LRT) REML with likelihood ratio test, REML (JK) REML with jackknife-based Wald test.

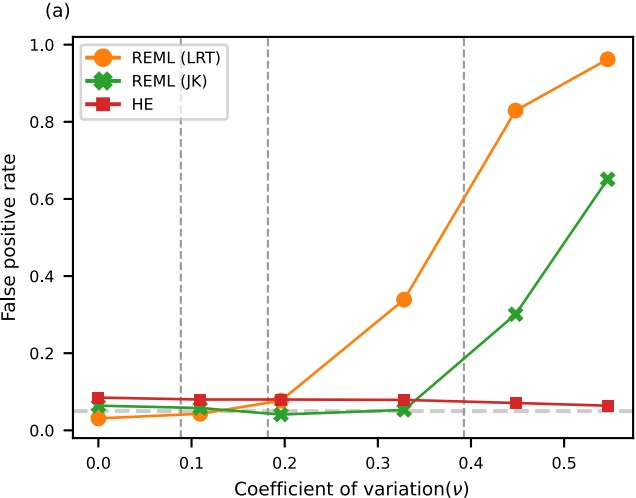

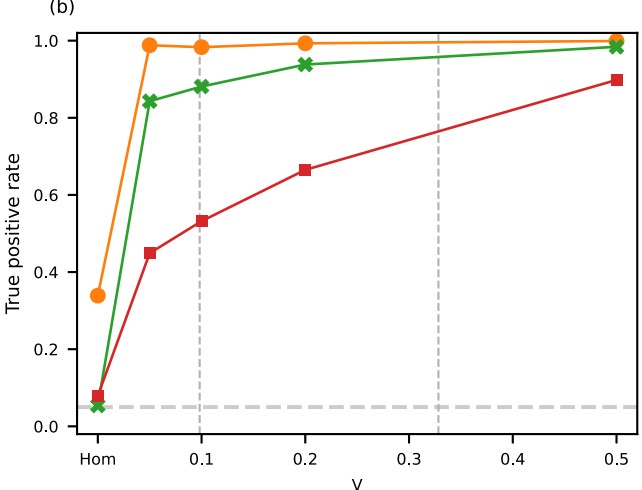

**Fig. 2 | Power of CTMM in simulations with uncertain estimates of noise variance ($\boldsymbol{\nu}_{ic}$). a** False positive rate under different levels of noise of $\boldsymbol{\nu}_{ic}$ in the simulation of the Hom model. Dashed lines indicate the 10, 50, and 90% percentiles of the transcriptome-wide distribution of coefficient of variation for $\nu_{ic}$ in the real iPSCs data; **b** True positive rate under noise of $\nu_{ic}$ with a coefficient of variation of 0.33 in the simulation of Free model, with varying cell type-specific variance for the first cell type from 0.05 to 0.5; cell type-specific variance for other three cell types

were fixed to 0.1; in Hom, all cell types had 0 cell type-specific variance. Dashed lines indicate the 10 and 50% percentiles of the transcriptome-wide distribution of cell type-specific variances in the real iPSCs data. A plot of false discovery rate and positive predictive value is shown in Supplementary Fig. 9 with a 50:50 mixture of genes simulated from the Hom and Free models. REML (LRT) REML with likelihood ratio test, REML (JK) REML with jackknife-based Wald test.

CTP, including REML (Wald), REML (JK), and HE, also had over 70% true positive rates when the sample size reached 100. ML and REML had similar performance when fitting CTP (Supplementary Fig. 6). We also assessed the impact of cell type proportions, number of cell types, and level of cell type-specific variance. As expected, the power increased when the main cell type became more common, when additional cell types were included, or when cell type-specific variance increased (Supplementary Fig. 6).

To examine the impact of uncertain estimates of $\boldsymbol{\nu}_{ic}$, we repeated the CTP simulation while incorporating noisy $\boldsymbol{\nu}_{ic}$. To be more realistic, this simulation was conducted with parameters estimated from real data of iPSCs differentiation. We first evaluated the uncertainty of $\boldsymbol{\nu}_{ic}$ by bootstrap resampling cells for all combinations of individual, cell type, and gene. Most of them had a coefficient of variation around 0.2 (Supplementary Fig. 7). To incorporate this uncertainty into the simulation, we added noise into $\nu_{ic}$ when fitting models (Methods). We tried five distributions of noise to cover the distribution of coefficient

of variation in real data (Supplementary Fig. 7). In the simulation of Hom model, when there was no noise of $\nu_{ic}$, that is a fitting model with real $\nu_{ic}$ used in simulations, REML (LRT) and REML (JK) were well calibrated, HE was slightly inflated (Fig. 2a). Along with the increase of noise, REML (LRT)'s false positive rate increased quickly and reached ~80% when using a high level of noise (coefficient of variation = 0.45); REML (JK) was rather resistant to noise that it only completely broke when unrealistically strong noise was added; while HE was not impacted by noise, it remained slightly inflated for all levels of noise. Of note, estimates of cell type-specific variance were weakly biased in REML under strong noise (Supplementary Fig. 8). In the simulation of the Free model, we found that REML (JK) had 80% of positive rate even when the cell type-specific variance is weak with 0.05 variance for the first cell type; HE also had intermediate power with about 50% of positive rate when the first cell type had 0.05 variance (Fig. 2b). Taken together, REML (JK) is the most powerful and robust method and is our primary approach in our iPSCs analysis.

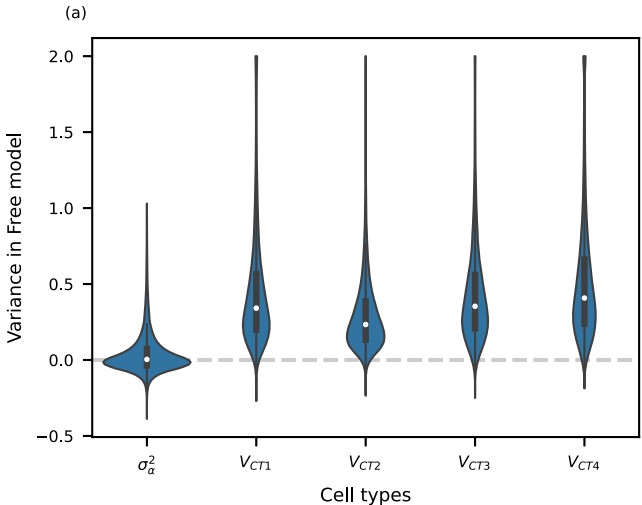

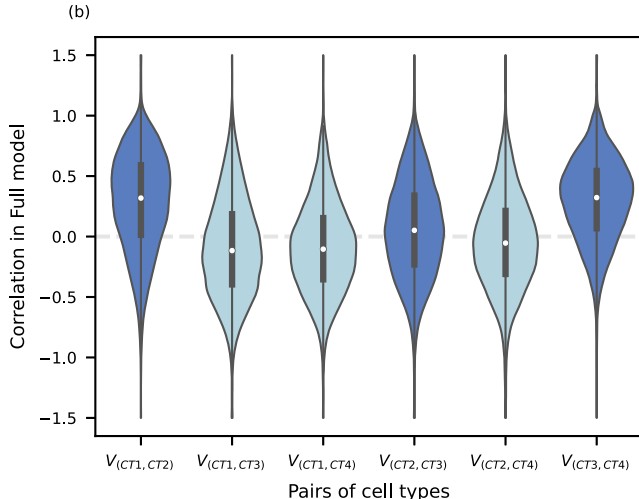

(a)

(b)

**Fig. 3 | Distribution of variance and correlation of cell type-specific effect across transcriptome from REML with CTP data. a** Homogeneous variance ($\sigma_\alpha^2$) shared across cell types and cell type-specific variance was assessed across 11,231 genes from the Free model; **b** Correlation of cell type-specific effect between cell types from the Full model, with dark blue indicating pairs of adjacent cell types and light blue indicating others. After removing genes with negative variances, 11,122

genes were left for correlation calculation. In **a**, values above 2 were truncated; in **b**, values above 1.5 or below −1.5 were truncated. About 109 genes were excluded from **b**, because of negative cell type-specific variance. Interior box plots show the median and the first and third quartiles. The whiskers extend to values within 1.5 times the interquartile range from the first and third quartiles. CT cell type.

Finally, we conducted simulations at the level of single cells to evaluate the impact of sequencing depth and the number of cells (Supplementary Note 1 Section 3.4). To assess whether our simulated count distribution is realistic, we compared it to the real data using countsimQC[23]. The comparison demonstrated a good fit to the real data in terms of the mean-variance distribution and the fraction of zeros per gene (Supplementary Fig. 10). We found that CTMM was robust across a realistic range for sequencing depth and number of cells (Supplementary Fig. 11A, C), though power improved with greater read depth or number of cells (Supplementary Fig. 11B, D).

**Application to human induced pluripotent stem cells**

We applied our methods to differentiating iPSCs[13]. Before fitting CTMM, we compared different approaches to imputing the cell type-specific pseudobulk ($\mathbf{y}_{ic}$, that is CTP). We evaluated single-gene imputation with softImpute and MVN and transcriptome-wide imputation with softImpute. We found that transcriptome-wide softImpute performed best (Supplementary Fig. 12A, C), though MVN performed similarly. We also compared approaches to impute the noise variance ($\mathbf{v}_{ic}$), which is required for CTMM. We observed similar results as for the pseudobulk in terms of mean squared error (Supplementary Fig. 12B, D). Based on these results, we used transcriptome-wide softImpute in practice.

We fit the Free model with both OP and CTP data using ML, REML, and HE (Supplementary Figs. 13, 14). We focus on REML with CTP, which was most powerful and robust in simulations. Transcriptome-wide, we found that the variation across individuals was almost entirely cell type-specific, as the homogeneous variance has a median close to 0 (median = 0.4%, Fig. 3a). By contrast, cell type-specific variance has a median of 32.8% across cell types. Weighted by cell type proportions, cell type-specific variation explained 14% of interindividual variation on average across the transcriptome (Supplementary Note 1 Eq. 3). Additionally, cell type proportion differences explained 12%, and residual cell-level variation ($\mathbf{v}$) explained 10%. The remaining variation is explained by covariates, especially PCs of pseudobulk gene expression (39%) and batch effects (21%). Note that cell subtype variation within an individual will be captured in the residual variation in $\mathbf{v}$, which also captures measurement errors (i.e., RNA transcripts that exist but are not sequenced), while interindividual variation in cell subtype proportions

will be captured in the interindividual covariance, $\mathbf{V}$. This illustrates the importance of modeling cell type-specific expression.

To evaluate the correlation of gene expression between cell types, we next fit the Full model. As expected, the correlations between adjacent development stages (CT1 and CT2, CT2 and CT3, and CT3 and CT4) were larger than the correlations between more distant stages (Fig. 3b). Furthermore, the correlation between CT2 and CT3 (median: 0.051) was smaller than the other adjacent stages (CT1-CT2 median: 0.318; CT3-CT4 median: 0.322). This is consistent with rapid changes in molecular profiles at day 2 (CT3)[13]. These patterns were also observed when fitting with ML or HE (Supplementary Fig. 13). When fitting OP, as expected, the estimates were far less precise, especially for HE (Supplementary Fig. 14).

Figure 4a shows gene expression differentiation in mean and variance in fitting CTP with REML (JK). We found many genes that were differentiated in variance between cell types, that is at least one cell type with non-zero cell type-specific variance. Among them, the top gene POU5F1 (Wald $p = 2.18 \times 10^{-27}$), also known as OCT4, is one of the three core transcription factors in the pluripotency gene regulatory network[24]. This signal was also confirmed in HE with CTP, where POU5F1 was the most significant signal in variance differentiation ($p = 1.33 \times 10^{-28}$, Supplementary Fig. 15). Although this gene was also significantly differentiated in mean, it is not outstanding in either REML or HE and less likely to be discovered for further functional analyses. To control for false positive, we identified candidate genes for cell type-specific variance as ordered by $p$ value in REML (JK) meanwhile requiring significant signals after Bonferroni correction in both REML (JK) and HE, with top 10 genes shown in Table 1. Among them, 85 genes were not differentiated in mean ($p > 0.01$ in REML), with some of those genes involved in processes like cell differentiation and growth (see top 10 of those genes in Table 2). Take NDUFB4 for example, there was no differentiation in mean between cell types ($p = 0.014$ in REML and $p = 0.125$ in HE), while significant differentiation in variance in both REML ($p = 1.93 \times 10^{-19}$) and HE ($p = 9.62 \times 10^{-15}$) (Fig. 5). We next performed GO enrichment analysis using clusterProfiler[25]. We tested the top 100 CTMM genes that did not have significant differences in mean ($p > 0.05$ after Bonferroni correction). We found dozens of significant enrichments, almost all of which reflect cellular metabolic activity (Supplementary Data 1). This finding aligns

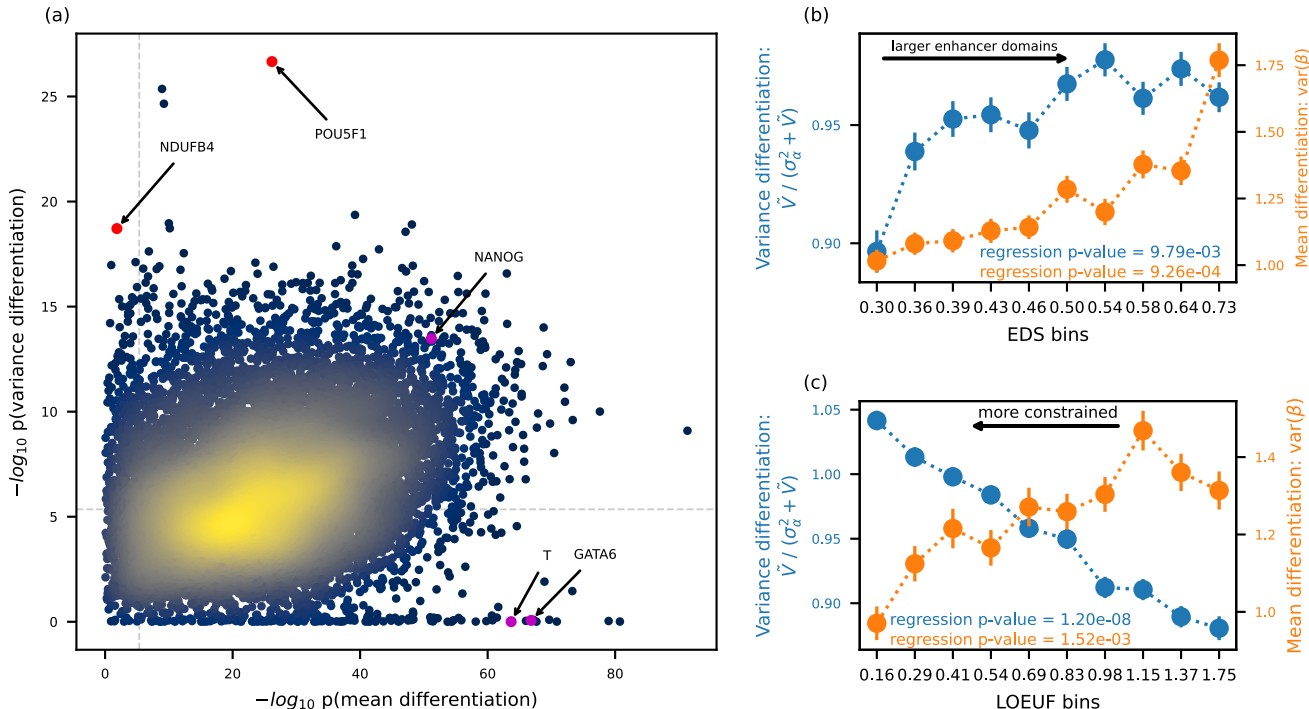

**Fig. 4 | Contrasting CTMM with ordinary differential expression transcriptome-wide. a** Each point conveys the cell type-specificity of a gene, where the x-axis tests for cell type-specific means (ordinary) while the y-axis tests for cell type-specific interindividual variance (CTMM). Genes described in the main text are highlighted, and Bonferroni-significance to adjust for multiple testing is illustrated as dashed horizontal and vertical lines. All results are from CTMM fit using REML and tested using the jackknife-based Wald test. **b, c** Plots quantify cell type-specificity of interindividual variance (blue, left) and mean expression (orange, right) across 10,891 genes compared to each decile in the transcriptome of EDS (**b**), or LOEUF (**c**). Each point is the average across all genes in that decile, and error bars display one standard error. *P* values correspond to the two-sided *t*-test for meta-regression of these mean points. The x-axis ticks show the median value of each feature in each decile.

**Table 1 | Top 10 genes significantly differentiated in variance between cell types in REML, while remaining significant in HE, with CTP data**

| Gene | REML | | HE | | Function |
|---|---|---|---|---|---|
| | *p*(variance)[a] | *p*(mean)[b] | *p*(variance) | *p*(mean) | |
| *POU5F1* | $2.18 \times 10^{-27}$ | $6.69 \times 10^{-27}$ | $1.33 \times 10^{-28}$ | $2.49 \times 10^{-23}$ | Cell pluripotency[24] |
| *HSPA8* | $4.32 \times 10^{-20}$ | $6.38 \times 10^{-40}$ | $1.51 \times 10^{-19}$ | $7.54 \times 10^{-42}$ | Cell pluripotency[46] |
| *SUB1* | $1.09 \times 10^{-19}$ | $1.00 \times 10^{-10}$ | $6.07 \times 10^{-16}$ | $1.03 \times 10^{-11}$ | Cell differentiation[47] |
| *NOP16* | $1.24 \times 10^{-19}$ | $7.66 \times 10^{-49}$ | $7.65 \times 10^{-15}$ | $4.74 \times 10^{-52}$ | Cell growth[48] |
| *RPL35* | $1.88 \times 10^{-19}$ | $7.17 \times 10^{-11}$ | $2.28 \times 10^{-20}$ | $5.70 \times 10^{-15}$ | Cell proliferation and survival[49] |
| *NDUFB4* | $1.93 \times 10^{-19}$ | 0.014 | $9.62 \times 10^{-15}$ | 0.125 | Cell differentiation[50] |
| *CCND1* | $2.75 \times 10^{-19}$ | $7.57 \times 10^{-48}$ | $4.15 \times 10^{-15}$ | $2.21 \times 10^{-44}$ | Cell differentiation[51] |
| *GYPC* | $1.34 \times 10^{-18}$ | $5.67 \times 10^{-37}$ | $1.10 \times 10^{-16}$ | $2.14 \times 10^{-27}$ | Cell differentiation[52] |
| *PTMA* | $1.89 \times 10^{-18}$ | $1.10 \times 10^{-43}$ | $1.60 \times 10^{-16}$ | $4.79 \times 10^{-40}$ | Apoptosis[53] |
| *SHFM1* | $2.36 \times 10^{-18}$ | $1.36 \times 10^{-7}$ | $1.20 \times 10^{-16}$ | $3.03 \times 10^{-5}$ | Cell pluripotency[54] |

[a]*p*(variance) indicates *p* values for variance differentiation between cell types;
[b]*p*(mean) indicates *p* values for mean differentiation between cell types. *P* values for testing mean differentiation and variance differentiation were calculated using the jackknife-based Wald test with CTP data from 94 individuals.

with the known importance of variation in metabolic state during iPSC differentiation[26]. Of note, there were three marker genes used in *Cuomo et al.*[13] to indicate each stem cell differentiation stage, spanning iPSC (*NANOG*), mesendoderm (*T*), and definitive endoderm (*GATA6*). We successfully detected significant mean differentiation in all three marker genes in both REML and HE; on the other hand, we detected significant variance differentiation in all three genes in HE, while only in the *NANOG* gene in REML, indicating loss of power in REML. We also note that mean differentiation had much stronger signals than

variance differentiation in both REML and HE (Supplementary Fig. 16). We compared *p* values for variance differentiation from different tests when fitting CTP (Supplementary Fig. 17). Generally, *p* values from different tests were largely consistent, except for REML with LRT. Specifically, for REML (JK) and HE, there were 4776 genes that were significant in both; 333 genes were significant only in HE, likely to be false positive; 2017 genes were significant only in REML (JK), partially due to false positive and partially due to higher power in REML (JK) than HE. We also conducted tests with OP data. Consistent with the low

**Table 2 | Top ten genes significantly differentiated between cell types in expression variance while not in mean ($p > 0.01$) in REML, meanwhile significantly differentiated in variance in HE, with CTP data**

| Gene | REML | | HE | | Function |
|---|---|---|---|---|---|
| | $p$(variance)[a] | $p$(mean)[b] | $p$(variance) | $p$(mean) | |
| NDUFB4 | $1.93 \times 10^{-19}$ | 0.014 | $9.62 \times 10^{-15}$ | 0.125 | Cell differentiation[50] |
| NUTF2 | $1.06 \times 10^{-17}$ | 0.109 | $1.16 \times 10^{-16}$ | 0.081 | Cell cycle[55], apoptosis[56] |
| SLX1A | $6.11 \times 10^{-15}$ | 0.162 | $7.17 \times 10^{-18}$ | 0.202 | Genome stability[57] |
| EIF4A1 | $2.29 \times 10^{-13}$ | 0.011 | $2.11 \times 10^{-10}$ | 0.698 | Stem cell self-renewal[58] |
| TFPI2 | $4.84 \times 10^{-13}$ | 0.024 | $7.30 \times 10^{-9}$ | 0.057 | Cell proliferation[59] |
| ATP5J2 | $6.30 \times 10^{-13}$ | 0.093 | $1.09 \times 10^{-13}$ | 0.012 | ATP synthesis[60] |
| SMAP1 | $1.02 \times 10^{-12}$ | 0.024 | $5.96 \times 10^{-10}$ | 0.049 | Cell growth[61] |
| NDN | $1.80 \times 10^{-12}$ | 0.011 | $8.21 \times 10^{-10}$ | 0.079 | Cell growth[62] |
| KRT10 | $2.40 \times 10^{-12}$ | 0.218 | $1.16 \times 10^{-10}$ | 0.403 | |
| NDUFA1 | $4.28 \times 10^{-12}$ | 0.406 | $2.64 \times 10^{-11}$ | 0.702 | Cell differentiation[50] |

[a]$p$(variance) indicates $p$ values for variance differentiation between cell types;
[b]$p$(mean) indicates $p$ values for mean differentiation between cell types. $P$ values for testing mean differentiation and variance differentiation were calculated using the jackknife-based Wald test with CTP data from 94 individuals.

power observed in simulations, we identified 23 genes that were significantly differentiated in variance in REML (LRT), and 0 genes were identified in HE (Supplementary Fig. 18).

## Gene features associated with cell type-specific variation

We next evaluated the relationship between CTMM results and four gene features related to genome structure and evolution. We compared CTMM's measure of cell type-specificity, which is based on interindividual variance, to a standard measure of cell type-specificity based on mean differences (Methods). First, we found that both CTMM and ordinary differential expression signals were enriched in genes with larger enhancer domains (based on the number of enhancers or enhancer domain score [EDS], Fig. 4b and Supplementary Fig. 19). These results align with previous findings that genes with larger enhancer domains were less likely to exhibit ubiquitous expression across tissues[27]. Second, we examined a measure of gene conservation called LOEUF (loss-of-function observed/expected upper bound fraction). We found that more-constrained genes have lower mean differences across cell types ($p = 1.0e-3$, Fig. 4c), consistent with previous findings that constrained genes were more frequently ubiquitously expressed across tissues[28,29]. CTMM's cell type-specificity measure also correlates with LOEUF, but in the opposite direction ($p = 1.2e-8$, Fig. 4c). Digging deeper, CTMM shows this primarily results from decreases in cell type-shared variation (Supplementary Fig. 19). In other words, stronger selection implies that cell types and individuals are more constrained toward their averages, so cell type-specific interindividual variation plays a larger role. We found qualitatively similar results using pLI (Supplementary Fig. 19).

## Application to peripheral blood mononuclear cells

The iPSC data[13] we analyzed above used plate-based sequencing. We next sought to confirm that CTMM is applicable to droplet-based sequencing, a different scRNA-seq technology that generally trades off a greater number of cells at the cost of lower read depth. We thus applied CTMM to the recent droplet-based data from the OneK1K cohort[22]. This dataset has many more individuals than the iPSC data

($N = 982$ vs 125) and more cells per individual (1300 vs 300), but it has far fewer reads per cell (~3 K vs ~0.5 M). The cells themselves are also very different, as OneK1K contains peripheral blood mononuclear cells (PBMCs) from living people rather than differentiating iPSCs from a controlled lab experiment. Another important difference is that the PBMC cell types are computationally inferred, while the iPSC cell types are defined by experimental days. Finally, the PBMC cell type proportions vary substantially (Supplementary Fig. 20). Our primary analysis was restricted to cell types with at least ten cells in at least 90% of individuals, resulting in seven cell types ($CD4_{NC}$, $CD4_{ET}$, $CD8_{ET}$, $CD8_{NC}$, NK, $B_{IN}$, and $B_{Mem}$, Supplementary Note 1 Section 4).

We find that CTMM provides powerful and robust estimates in the OneK1K data. First, CTMM detected significant cell type-specific interindividual variance for 2310 genes out of the 11,526 total genes tested ($p < 0.05/11526$, Supplementary Fig. 21). The top signal is $RPS26$ ($p = 1.78 \times 10^{-167}$, Supplementary Fig. 22), which plays a key role in regulating T cells[30,31]. Further, genetic variation causes interindividual variation in this gene that is cell type-specific[22] and is linked to complex traits such as eczema and asthma[32]. As in the iPSCs, we tested GO enrichment in the top CTMM genes. In this large dataset, almost all genes have significant differential mean expression, so we tested the top 100 CTMM genes irrespective of their mean differences. Almost all of the top enrichments relate to immune function, including several that are specific to leukocytes (Supplementary Data 2). Second, CTMM partitions transcriptome-wide interindividual variation into components that are shared across cell types (10.9%) vs cell type-specific (21.5% on average across cell types, Supplementary Fig. 23). This is an interesting contrast with the iPSC results, where shared interindividual variation was near zero. Biologically, this could be explained by differences in cellular environment: individual-level covariates like age, smoking, or BMI may have shared effects across cell types and they are likely to have larger effects on PBMCs in whole blood than iPSCs in a controlled lab. Finally, we evaluated CTMM's Full model transcriptome-wide to quantify interindividual covariance between cell types. These estimates recapitulated expected relationships between cell types (Supplementary Fig. 24). For example, the most-correlated cell types are $CD4_{NC}$ and $CD4_{ET}$; intuitively, this means that an individual with an above-average expression of a gene in their $CD4_{NC}$ cells will typically also have an above-average expression in their $CD4_{ET}$ cells. The second most-correlated cell types are $CD4_{NC}$ and $CD8_{NC}$, which is consistent with the observation that $CD4_{NC}$ and $CD8_{NC}$ shared the most genetic effects in prior work[22].

We next tested the robustness of CTMM to rarer cell types in a secondary analysis that includes two additional cell types, $Mono_C$ and $CD8_{S100B}$. CTMM gave consistent results for the seven larger cell types that are included in both analyses (Supplementary Figs. 25–29). As expected, CTMM's estimates are noisier for $Mono_C$ and $CD8_{S100B}$, which are rarer cell types. Nonetheless, adding these cell types enables CTMM to discover new differentially-variable genes. For example, the top newly-significant gene is $TMEM176B$ ($p = 5.45 \times 10^{-63}$ vs $p = 0.18$ in our primary analysis), which makes sense as this gene is primarily expressed in $Mono_C$ (Supplementary Fig. 30). We conclude that CTMM's results are robust to variations in the input cell types, but its estimates are less accurate for rarer cell types.

## Discussion

Mean differences in gene expression across cell types are well documented and are the primary focus of most scRNA-seq analyses. Here, we have introduced a new model called CTMM to quantify variance differences across cell types in scRNA-seq data. Bulk expression analyses have established that interindividual variance in expression can be important for characterizing disease biology[33] and identifying context-dependent genetic effects[34]. The key innovation in CTMM is adapting Gaussian LMMs to scRNA-seq data, which is challenging because scRNA-seq data are highly noisy and non-Gaussian. The key

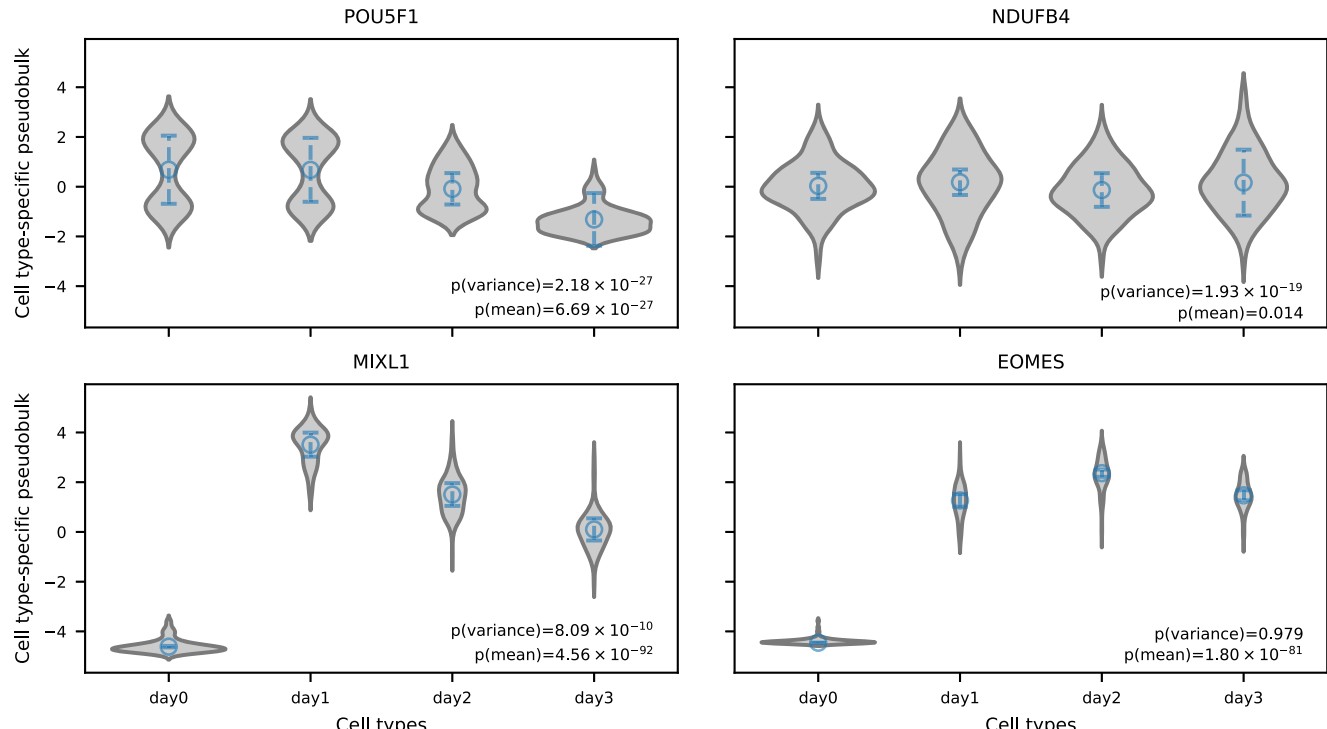

**Fig. 5 | Estimates of cell type fixed effect and cell type-specific variance for specific genes in REML with CTP data.** Four genes were chosen as examples. *POU5F1* exhibited the strongest signal of variance differentiation among all genes; *NDUFB4* exhibited the strongest signal of variance differentiation among genes without signals of mean differentiation (*p*>0.01); *MIXL1* exhibited the strongest signal of mean differentiation among all genes; and *EOMES* exhibited the strongest signal of mean differentiation among genes without signals of variance differentiation (*p*>0.01). The violin plot represents the distribution of cell type-specific pseudobulk after standardizing overall pseudobulk to mean 0 and variance 1; circles indicate estimated cell type fixed effects; the length of the dashed line indicates twice the sum of homogeneous variance shared across cell types and cell type-specific variance. *P* values for testing mean differentiation and variance differentiation were calculated using the jackknife-based Wald test with CTP data from 94 individuals.

idea is to summarize the scRNA-seq data into cell type-specific pseudobulk[21], which enables approximately unbiased inference with CTMM on as few as 20 individuals. We carefully profile several standard methods to fit LMMs and propose a jackknife-based test using REML as the most powerful and robust method, which we support with extensive simulations and analyses of differentiating iPSCs and PBMCs. We implement and freely release these methods as a user-friendly Python package. We expect that CTMM will be an important step toward robust and rich variance decompositions of scRNA-seq data, which will be increasingly powerful and informative as scRNA-seq sample sizes grow.

In the limiting case with infinite cells, when the measurement error is reduced to 0, CTMM simplifies to a typical LMM on bulk expression. In this case, CTMM with Overall Pseudobulk (OP) data were comparable to decomposing variance in bulk expression data using computationally deconvolved cell type proportions. The significant benefit is that scRNA data provides much better estimates of cell type proportions, which can both reduce false positives and improve power. Likewise, CTMM with Cell Type-specific Pseudobulk (CTP) data becomes comparable to bulk analyses of sorted cells without the need for sorting pre-defined cell types. In practice, when the number of cells is limited, another significant benefit of CTMM over bulk analyses is the ability to distinguish biological variance across individuals from measurement error, which is especially important when measurement error varies across individuals, cell types, or experimental conditions. However, the disadvantage of CTMM compared to bulk is that it requires larger sample sizes, which is currently expensive.

CTMM has several important limitations. First, as scRNA data in individual cells is highly non-Gaussian, CTMM's Gaussian assumption relies on combining many cells and the central limit theorem. In

practice, we require >10 cells per individual-cell type pair, which limits CTMM to common cell types. A related concern is that lowly-expressed genes can be severely non-Gaussian, increasing the number of cells needed for the Gaussian approximation. We find that higher overall levels of a gene's expression increase the power of CTMM (Supplementary Fig. 31), which also holds for most tests of scRNA-seq data. Second, CTMM assumes cell types are already known. Our iPSCs data analysis solves this by defining cell types based on experimental days. However, most studies infer cell types directly from the scRNA-seq data, such as the cell types in our PMBC data analysis, inducing some circularity; this is typically ignored[2,7,35] yet will deflate estimates of cell type-specific variance by construction. Third, CTMM assumes discrete cell types, whereas continuous cell types are more appropriate in some cases, e.g., when defined by pseudo-time[13,20,36] or degree of IFN stimulation[21]. While incorporating continuous cell types is straightforward with overall pseudobulk data, it can only be expressed in cell type-specific pseudobulk data by discretizing the continuous cell types. Fourth, it is well-known that count data evince a complex mean-variance relationship, and studies have observed that the variance of gene expression across cells is dependent on mean expression[37]. Nonetheless, simulations show that this problem is unlikely to be important in practice (Supplementary Fig. 32). Moreover, we find biologically plausible genes with significant differential variance but without significant differential mean, showing that modeling variance has utility beyond merely tagging mean signals. Fifth, CTMM only considers pseudobulk data, which greatly improves computational efficiency and facilitates its simplifying Gaussian assumptions. Nonetheless, pseudobulk inherently discards cell-level information, which sacrifices statistical power and resolution within cell types. Moreover, cell type-specific pseudobulk necessarily discretized cells into

categories, which can be somewhat arbitrary. Finally, despite our use of careful nonparametric tests, our Free test for cell type-specificity remains slightly inflated, emphasizing the importance of biologically validating and replicating results. While this inflation is small for sample sizes around ~100 and vanishes for sample sizes above ~300, CTMM is not reliable for sample sizes below ~50. Nonetheless, CTMM's estimates remain unbiased (Supplementary Fig. 1), hence it can be used to profile transcriptome-wide averages for any sample size. Also, we developed a simplified version of CTMM which remains calibrated for sample sizes ~50, but it assumes that all cell type-specific variances are equal (Supplementary Fig. 33).

CTMM is a step toward translating well-established LMM methodologies to scRNA-seq data. A key extension of CTMM is to quantify cell type-specific heritability of gene expression, which is typically more powerful than single-SNP tests of context-specific genetic regulation[12,21]. Because CTMM models cell-level noise, it can eliminate downward biases in heritability that are unavoidable in bulk expression data. Another extension is to jointly model covariance across both cell types and genes. For example, this enables identifying cell type-shared and -specific networks. This, too, is necessarily biased in bulk expression data, where covarying measurement errors will confound biologically meaningful networks. The Full model can also be extended to learn structured networks between cell types by leveraging penalized covariance estimates[38] or by specifically tailoring it to a given application; for example, we could restrict $V$ to be banded to capture temporal structure in the differentiating iPSCs. It would also be useful to extend CTMM to test for variance differences between groups of individuals, e.g., disease cases and controls. Nonetheless, this will require careful modeling and robustness tests to account for subtle ascertainment biases. The long-term goal is to combine together these features into a comprehensive model of transcriptomic covariation across cells, cell types, individuals, and environments in order to understand genetic and nongenetic drivers of complex diseases. Overall, we consider CTMM an important step on a long path to fully understanding the causes and consequences of variation within and between individuals in scRNA-seq data.

## Methods
### Models
**Overview of cell type-specific linear mixed models for gene expression.** We model the expression level of a given gene for individual $i$, cell type $c$, and cell $s$ by:

$$y_{ics} = \boldsymbol{\beta}_c + \boldsymbol{\alpha}_i + \boldsymbol{\Gamma}_{ic} + \epsilon_{ics} \tag{3}$$

In this model, $y_{ics}$ is the gene expression level for the $s$-th cell from cell type $c$ in individual $i$; note that the number of measured cells varies across individuals and cell types. $\boldsymbol{\beta}_c$ is the mean expression level in cell type $c$, which we model as a fixed effect. $\boldsymbol{\alpha}_i$ captures differences between individuals that are shared across cell types, which we model as a random effect: $\boldsymbol{\alpha}_i \sim^{iid} N(0, \sigma_\alpha^2)$. $\boldsymbol{\Gamma}_{ic}$ captures the difference between individuals that is specific to cell type $c$, which we also model as a random effect by $\boldsymbol{\Gamma}_{i,} \sim^{iid} N(0, \mathbf{V})$. Here $\boldsymbol{\Gamma}_{i,}$ is a vector of cell type-specific expression for individual $i$ and $\mathbf{V}$ is a $C \times C$ matrix describing cell type-specific variances and covariances across cell types.

Finally, $\epsilon_{ics}$ is the residual effect, which we assume to be *i.i.d.* for each individual-cell type pair with $E(\epsilon_{ics}) = 0$ and $V(\epsilon_{ics}) = \sigma_{ic}^2$. We directly estimate $\sigma_{ic}^2$ from the single cell-level data by $\hat{\sigma}_{ic}^2 := \sum_{s=1}^{n_{ic}} (y_{ics} - y_{ic})^2 / (n_{ic} - 1)$, where $n_{ic}$ is the number of cells in individual $i$ and cell type $c$ and $y_{ic}$ is the average expression across all $n_{ic}$ cells (*i.e.*, cell type-specific pseudobulk, defined below). $\hat{\sigma}_{ic}^2$ is unbiased even if $\epsilon_{ics}$ is non-Gaussian, which is important because expression in single cells is non-Gaussian. Note that this is impossible in bulk expression data, even if sorted into cell types, because bulk only measures average expression. That is, scRNA-seq data makes it

possible to distinguish true interindividual variation from measurement noise.

Our focus is the covariance matrix $\mathbf{V}$, which captures the differences and similarities between cell types (for a given gene). The diagonal terms capture cell type-specific variance. If there is no cell type-specific variation between individuals, then $\mathbf{V}_{cc} = 0$ for all $c$. The off-diagonal terms capture covariance between specific pairs of cell types; if all cell types are equally similar to each other, then $\mathbf{V}_{cc'} = 0$ for all $c \neq c'$.

We consider three nested models of interindividual variation defined by the structure of $\mathbf{V}$. First, the homogeneous (Hom) model assumes that $\mathbf{V} = 0$, i.e., that all expression variance is shared homogeneously across cell types without any cell type-specificity. Second, the Free model allows arbitrary levels of cell type-specific variance by allowing $\mathbf{V}$ to be an arbitrary diagonal matrix. Third, the Full model captures arbitrary levels of covariance between specific cell type pairs by allowing $\mathbf{V}$ to be any positive semidefinite matrix. Intuitively, the Hom model captures variation across individuals, but assumes this variation is identically shared across cell types. The Free model allows cell type-specific variation, *e.g.*, a gene that is largely similar between individuals except in a single cell type. The Full models allows complex relationships among cell types, e.g., hierarchical relationships among immune cell types.

A technical consideration in the Full model is that $\mathbf{V}$ and $\sigma_\alpha^2$ are not jointly identified. Specifically, passing a constant between $\sigma_\alpha^2$ and $\mathbf{V}$ does not change the likelihood (i.e., $L(\sigma_\alpha^2, \mathbf{V}) \equiv L(\sigma_\alpha^2 - \lambda, \mathbf{V} + \lambda \mathbf{J}_C)$, where $\mathbf{J}_C$ is $C \times C$ matrix containing all 1s). Therefore, without loss of generality, we set $\sigma_\alpha^2 = 0$ in the Full model. The Full model is statistically challenging because its number of parameters scales quadratically with the number of cell types, $C$. In practice, the Full model only has precise estimates with hundreds to thousands of samples or, as below, when aggregating together many genes.

**Deriving models for overall and cell type-specific pseudobulk.** Directly modeling single cell expression as in Eq. 3 is challenging computationally and statistically. Computationally, modeling individual cells increases the number of observations by orders of magnitude because there can be dozens or hundreds of cells per individual-cell type pair. Statistically, the individual cell's expression is highly non-Gaussian, requiring additional assumptions and computationally expensive generalized linear mixed models. Instead, we study scRNA-seq data at the level of pseudobulk expression, which averages expression over many cells. We consider both overall pseudobulk (OP), which averages over all measured cells per individual, and cell type-specific pseudobulk (CTP), which averages over cells in each cell type per individual.

Specifically, the pseudobulk measures that we input to CTMM are:

$$\mathbf{y}_i := \frac{1}{n_i} \sum_{c=1}^{C} \sum_{s=1}^{n_{ic}} y_{ics} \quad \text{and} \quad \mathbf{y}_{ic} := \frac{1}{n_{ic}} \sum_{s=1}^{n_{ic}} y_{ics} \tag{4}$$

where $\mathbf{y}_i$ is the OP expression for individual $i$, and $\mathbf{y}_{ic}$ is the CTP expression for individual $i$ and cell type $c$.

Our cell-level model in Eq. 3 implies the following mixed model for the OP expression:

$$\mathbf{y}_i = \underbrace{\sum_c \mathbf{P}_{ic} \boldsymbol{\beta}_c}_{\substack{\text{cell type–specific} \\ \text{mean}}} + \underbrace{\boldsymbol{\alpha}_i}_{\substack{\text{cell type–shared} \\ \text{variation}}} + \underbrace{\sum_c \mathbf{P}_{ic} \boldsymbol{\Gamma}_{ic}}_{\substack{\text{cell type–specific} \\ \text{variation}}} + \underbrace{\boldsymbol{\delta}_i}_{\substack{\text{measurement} \\ \text{noise}}}$$

$$\tag{5}$$

with $\boldsymbol{\delta}_i := \frac{1}{n_i} \sum_{c=1}^{C} \sum_{s=1}^{n_{ic}} \epsilon_{ics} \sim^{ind} N(0, \boldsymbol{v}_i)$; $\boldsymbol{v}_i := \sum_{c=1}^{C} \frac{n_{ic}}{n_i^2} \sigma_{ic}^2$

$\boldsymbol{\delta}_i$ is the measurement noise for individual $i$, with variance $\boldsymbol{v}_i$ that we estimate by plugging in our estimate of $\sigma_{ic}^2$. $\mathbf{P}$ is the matrix of cell type proportions, defined by $\mathbf{P}_{ic} := \frac{n_{ic}}{n_i}$.

Our cell-level model in Eq. 3 also implies a mixed model on the CTP expression data:

$$\mathbf{y}_{ic} = \boldsymbol{\beta}_c + \boldsymbol{\alpha}_i + \boldsymbol{\Gamma}_{ic} + \boldsymbol{\delta}_{ic} \qquad (6)$$

with $\boldsymbol{\delta}_{ic} := \frac{1}{n_{ic}}\sum_{s=1}^{n_{ic}}\epsilon_{ics} \sim^{ind} N(0, \mathbf{v}_{ic}); \mathbf{v}_{ic} = \frac{\sigma_{ic}^2}{n_{ic}}$

Here, $\boldsymbol{\delta}_{ic}$ is the noise for individual $i$ and cell type $c$, with variance $\mathbf{v}_{ic}$. By the central limit theorem, both $\boldsymbol{\delta}_i$ and $\boldsymbol{\delta}_{ic}$ are approximately Gaussian when $n_{ic}$ is not too small, even though $\epsilon_{ics}$ is very non-Gaussian. Note that Eq. 6 is the same as Eq. 2.

## Fitting and testing parameters of CTMM

We evaluated three approaches to estimate the parameters in CTMM: ML, REML, and HE. We implemented ML by maximizing the likelihood function using the BFGS algorithm implemented in the R function `optim' (Supplementary Note 1). REML was fit similarly using the restricted likelihood, which residualizes covariates from the full likelihood (Supplementary Note 1). For both REML and ML, we reran 10 random restarts if the initial optimization attempt failed (Supplementary Note 1), which is important to mitigate bias from local maxima with modest sample sizes. We allowed negative variance components to reduce bias, though the total expression variance was always positive. Due to the complexity of these likelihood functions, we evaluated refining the BFGS solution with Nelder-Mead iterations; we found that this is not necessary for the analyses considered in the Main text, but it can be important for the more challenging analyses, such as fitting the Free model with ML on OP data (Supplementary Fig. 34). We fit HE analytically (Supplementary Note 1).

Because the CTP expression data is a vector of length $NC$, where $N$ is the number of individuals, naively fitting CTMM in ML and REML has a computational complexity of $O(N^3C^3)$. We use several linear algebra identities to simplify the complexity to $O(NC^3)$. The relative gains will increase as $N$ and $C$ grow, which are both expected in future scRNA-seq datasets.

Our primary test compared the Hom and Free models, which asks whether interindividual variation is cell type-specific or shared uniformly across cell types. In other words, this is a test for differential expression variance across cell types. By comparison, standard tests for differential expression ask whether the mean expression levels, $\boldsymbol{\beta}_c$, differ across cell types.

We implemented LRT and Wald tests to compare the Free and Hom models. For the LRT, we used $C$ degrees of freedom because the Free model adds variance components for each cell type (but see ref. 39 and ref. 40 for a more detailed discussion of these tests). For the Wald test, we used an $F$-test with $C$ numerator degrees of freedom and $N - R$ denominator degrees of freedom, where $R$ is the number of model parameters in the Free model, that is $2C + 1$. We evaluated two options to estimate the precision matrix for CTMM's variance component estimates, which is needed for the Wald test. First, we used the inverse of the Fisher information matrix for REML and ML, which is consistent for large sample sizes. Second, we used a jackknife to nonparametrically estimate the precision matrix by fitting the model after excluding each sample in turn. For large sample sizes, both LRT and Wald tests are valid; however, we are interested in modest sample sizes, and hence, we profile a wide range of approaches.

We tested for mean expression differentiation by evaluating the null hypothesis that $\boldsymbol{\beta}_c = \boldsymbol{\beta}_{c'}$ for all cell types $c$ and $c'$. $\boldsymbol{\beta}$ is the cell type fixed effect (i.e., cell type-specific mean expression) and is estimated using generalized least squares with variance components fit under the Free model. We used a Wald test for $\boldsymbol{\beta}$ with numerator degrees of freedom $C - 1$ and denominator degrees of freedom $N - R$, where $R$ is the number of parameters in the Free model, that is $2C + 1$. We estimated the precision matrix for CTMM's estimates of $\boldsymbol{\beta}$ using jackknife. The jackknife includes re-fitting variance components, which is important because these variance component estimates are noisy. As

the covariance matrix estimated by HE can be singular and the fixed effects are not our focus, we tested for mean expression differentiation simply using ordinary least squares.

We have also extended CTMM to accommodate additional random effects (Supplementary Note 1). This can be essential in practice, but it can be computationally infeasible as it requires inverting large matrices. Fortunately, the primary use case involves blocked random effects, such as experimental batch in our iPSC analysis. We derived a different optimization approach designed specifically for this common scenario, which simplified the computational complexity by orders of magnitude. In our iPSC analysis, these manipulations reduced REML computation time per gene from ~40 to ~10 s.

## Prior LMM applications to scRNA-seq data

LMMs are a basic statistical framework for partitioning variation, and several prior studies have applied LMMs to scRNA-seq data (Supplementary Table 1). Most prior work has applied generic LMM methods at the level of single cells. They fit variance components for batch effects[11,13], experimental context[10], and/or some form of interindividual variation[11–14]. Additionally, some of these studies model the non-Gaussianity of cell-level expression data[10,14]. Despite the strengths of these works, none aim to partition shared vs specific components of interindividual variation. This is the key novelty in CTMM, and it requires a different variance component model than the ones that have been used in prior work. At a more technical level, CTMM develops a novel testing framework based on jackknife rather than use off-the-shelf tools, which solves biases in standard LMM variance component tests due to the complexity of scRNA-seq data.

## Simulation

We tested the performance of CTMM with a series of simulations under Hom and Free models. We simulated gene expression for each individual from Eq. 5 (for overall pseudobulk) and Eq. 6 (for cell type-specific pseudobulk). We varied the number of individuals, cell type proportions, number of cell types, and levels of cell type-specific variance. For each simulated dataset, we evaluated all three methods to fit CTMM (ML, REML, and HE) and each applicable test for cell type-specific interindividual variance (LRT and Wald). For each simulation parameter setting, we ran 1000 replicate simulations to calculate the average CTMM estimates, their sampling distributions, and the test positive rate. We also simulated under the more complex Full model. Further simulation details are provided in Supplementary Note 1, with a list of simulation parameters in Supplementary Table 2.

We also performed simulations to assess CTMM's sensitivity to estimation errors in $\nu_{ic}$, the level of noise due to cell-level variation. This is important because $\nu_{ic}$ is not known in practice. Specifically, for each $\mathbf{v}_{ic}$, we draw $x_{ic}$ i.i.d. from a $Beta(2, b)$ distribution and then add $+x_{ic}\mathbf{v}_{ic}$ or $-x_{ic}\mathbf{v}_{ic}$ before inputting $\mathbf{v}_{ic}$ to CTMM (Supplementary Note 1). To span the range of estimation errors in the real iPSCs data, we simulated $b = 20, 10, 5, 3, 2$. To evaluate power under a range of Free models, we varied cell type-specific variance for cell type 1 ($\mathbf{V}_{11}$) from 0.05 to 0.5 and fixed other cell type-specific variances to 0.1. For simplicity, the Free model simulations are always used $b = 5$ (the most realistic value). As this simulation focuses on CTMM's utility in our real data analysis, we simulated using the parameters we estimated in the iPSCs data below (Supplementary Note 1), and we only examined CTP as it is far more powerful. We ran 1000 replicates for each setting of simulation parameters.

## Differentiating iPSCs analysis

**Data and model.** Human induced pluripotent stem cells (iPSCs) are derived from somatic cells that have been reprogrammed into an embryonic-like pluripotent state. iPSCs can differentiate into diverse cell types, with a concomitant transcriptomic trajectory across time as the cells differentiate. We studied the transcriptome as iPSCs

differentiate into endoderm using scRNA-seq data from 125 individual donors[13]. Cells were collected on four consecutive days as the iPSCs differentiated, starting from iPSCs, which we used to define four cell types. We used the log-transformed gene expression data provided by Cuomo et al., which has been through a thorough process of quality control and normalization (https://zenodo.org/record/3625024#.Xil-0y2cZ0s). The dataset includes 11,231 genes and 36,044 cells.

For the 33 individuals who had technical replicates in the data, we only included the replicates with the largest number of cells. We excluded individuals with fewer than 100 cells to better satisfy the Gaussian approximation of $\delta_{ic}$, leaving 94 individuals.

For each gene, we standardized OP and CTP by scaling such that OP has a mean of 0 and a variance of 1. We then fit this scaled OP and CTP expression into Hom, Free, and Full models with ML, REML, and HE. In all models, we adjusted for sex, neonatal diabetes, and the first six principal components calculated on OP expression as fixed effects. We used our extension of CTMM to model the experimental batch as a random effect, which is important because the batch has large effects that cannot be ignored yet has too many degrees of freedom to fit as fixed effects (24 batches vs 94 individuals). We used Bonferroni correction to account for multiple testing across genes.

**Impute pseudobulk data.** For individual-cell type pairs with no more than ten cells, $\mathbf{y}_{ic}$ and $\mathbf{v}_{ic}$ were set to missing. Requiring more than ten cells is our default guidance (in practice, we find that our results are robust to modifying this cutoff from ten cells to 5 or 20, Supplementary Fig. 35). We then imputed missing entries in $\mathbf{y}_{ic}$ and $\mathbf{v}_{ic}$. We compared three approaches to imputation (each applied separately to $\mathbf{y}$ and $\mathbf{v}$). First, we imputed each gene separately using either softImpute[41] or MVN-impute (implemented in ref. [42]). In brief, the former makes a low-rank approximation, while the latter approximates individuals as independent and leverages correlations among cell types. We also evaluated imputing all genes jointly across the transcriptome using softImpute (in an $N \times CG$ matrix, where $G$ is the number of genes); this is computationally impossible with MVN-impute.

To evaluate imputation accuracy, we masked observed entries in $\mathbf{y}_{ic}$ and $\mathbf{v}_{ic}$ and compared the imputed values to the masked true values. Of note, if one cell type of an individual has less than or equal to ten cells, all genes' expression would be missing for the pair of individual and cell type. To be realistic, we maintained this structure of missingness by employing a "copy-mask" approach as in our prior study[42]. We randomly sampled an individual with missing cell types and masked the same cell types in another randomly chosen individual. We repeated this process until 10% of all pairs of individual and cell type were masked. We calculated correlation and mean squared error (MSE) between imputed values and masked true values across individuals for each gene-cell type pair. We conducted ten replications of the process of masking and imputation and calculated the medians of correlation and MSE across those repeats as final measures of imputation accuracy. For $\mathbf{v}_{ic}$, imputation might get negative values by chance. We treated these negative variances in different ways in OP and CTP expression data. In OP, we set negative variances to 0, so they had little impact on the estimation of $\mathbf{v}_i$ while maintaining information from other cell types; in CTP, for each gene and cell type, we set them to maximum raw $\mathbf{v}_{ic}$ in that specific gene and cell type, so they contributed less to model likelihood. Note that standard approaches to impute expression in single cells[43,44] does not impute the pseudobulk data, which has missing entries due to missing cells, not missing expression within observed cells.

**Enrichment of gene features related to enhancers and selection.** We evaluated four gene-level features: LOEUF, pLI, EDS, and the number of enhancers. LOEUF and pLI were obtained from the Genome Aggregation Database (gnomAD) version 2.1[29]. LOEUF and pLI measure a gene's susceptibility to loss-of-function mutations, and they approximately quantify the degree of selection on a gene. EDS and the number of enhancers were obtained from Wang and Goldstein[27]. The number of enhancers was computed from enhancer-gene links inferred by ref. [45] based on chromatin state and correlation of histone modifications with gene expression. EDS is a comprehensive score derived from 108 features associated with enhancer domains, including the number of enhancers. It reflects the size and redundancy of enhancer domains in a gene.

For each feature, genes were stratified into deciles based on their respective feature scores. Subsequently, we computed both the mean and median values of various gene expression properties within each decile, as well as their standard errors. These gene expression properties are:

- the total interindividual variance, which sums the cell type-shared variance with the average cell type-specific variance: $\sigma_\alpha^2 + \widetilde{V}$, where $\widetilde{V} = \frac{1}{C}\sum_{c=1}^{C}\mathbf{V}_{cc}$ is the average cell type-specific variance.
- the proportion of interindividual variation that is cell type-specific, defined by $\frac{\widetilde{V}}{\sigma_\alpha^2 + \widetilde{V}}$.
- the amount of mean differences across cell types, quantified by the variance of the mean expression level across cell types: $\mathrm{var}(\beta)$.
- the positive rate for two cell type-specificity tests: CTMM's test of cell type-specific interindividual variance and the ordinary test of cell type-specific mean expression.

To robustly examine the broad relationship between CTMM results and gene features, we performed a meta-regression of each decile's mean and median CTMM results against the decile index using ordinary least squares.

### Reporting summary
Further information on research design is available in the Nature Portfolio Reporting Summary linked to this article.

## Data availability
Processed single cell count data from iPSCs are publicly available from Zenodo: https://zenodo.org/record/3625024#.Xil-0y2cZ0s. OneK1K single-cell gene expression data are publicly available via Gene Expression Omnibus (GSE196830 [https://www.ncbi.nlm.nih.gov/geo/query/acc.cgi?acc=GSE196830]). The simulated datasets and imputed pseudobulk data were fully reproducible using the code provided in the study and the publicly available iPSCs and OneK1K data. All data generated during this study are included in this published article and its supplementary information files.

## Code availability
The CTMM Python package, along with Python (version 3.11.5) and R (version 4.3.1) code used for all analyses in this paper, is available at: https://github.com/Minhui-Chen/CTMM.

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

## Acknowledgements

This work was funded by K25HL157603 (to A.D.). We thank Joseph E. Powell for kindly providing a post-quality control version of the OneK1K data. We thank the Center for Research Informatics and the Research Computing Center for providing the computing resources. The Center for Research Informatics is funded by the Biological Sciences Division at the University of Chicago with additional funding provided by the Institute for Translational Medicine, CTSA grant number 2U54TR002389-06 from the National Institutes of Health. We thank Ben Umans for helpful feedback and Xuanyao Liu for feedback and help in defining gene features related to enhancers and selection.

## Author contributions

M.C. developed statistical methodology, performed analyses, and wrote the manuscript. A.D. conceived and supervised the project and wrote the manuscript.

## Competing interests

The authors declare no competing interests.
