## [Peer Review File · Nature Communications]

A robust model for cell type-specific interindividual variation in single-cell RNA sequencing dataReviewer #1 (Remarks to the Author):

Chen and Dahl propose CTMM, a linear mixed model framework to test for cell type-specific intra-individual variation in single-cell data from multiple individuals. Overall I think this model is well presented and robust, and a useful addition to the field.

My main concern is around performance in common droplet-based single-cell data, both in terms of the distribution of (pseudobulk) counts, and scalability. I also believe the author may have overstated the method's novelty a little.

Main Comment

The authors use both simulated and real data to evaluate performance of the method. In the simulation experiments, synthetic data was generated from the models themselves, and a few parameter values were estimated. While I appreciate that not every single parameter value can be tested, and that some effort was put into making the simulations more realistic by estimating values from the iPSC data, I think there are a few aspects that should be taken into consideration. First, the choice of using 4 cell types seems arbitrary, and particularly low. I would argue most scRNA-seq studies identify 10+ clusters (see also my comment below on discrete cell types), so performance should be evaluated for a higher number of cell subpopulations (types). Second, the authors repeatedly mention the small sample sizes in single-cell data. While this is absolutely true compared to population genetic studies, there are now single-cell datasets from several hundreds of individuals – e.g., Yazar et al, ref. 18 includes nearly 1,000 individuals. How does CTMM perform on such sample sizes? The model should be tested on a larger dataset, or at least simulations should include sample sizes up to 1,000. Third, I may have missed it, but was the number of cells per individual varied when estimating pseudobulk? In real settings this varies widely and it would be good to evaluate the model in different circumstances. Also, why was a single (randomly drawn) gene used for more realistic simulations? What was its distribution? At least a few genes should be considered here, chosen at different overall mean and variance levels. Finally, and in relation to the real data used. Cuomo et al use plate-based scRNA-seq data (SmartSeq2), which behave quite differently to droplet-based data, which is by far more popular today. In particular, this is full-length data and results in more reads per cells, but fewer cells in general (just over 36,000 cells, compared to well over a million in e.g., refs 16-18). I think performance on e.g., 10X scRNA-seq data (for example, refs. 16-18) is needed and would ensure the method will be more generalisable to real settings.

Other Comments

Can the authors comment on the genes that were found to be significantly differentiated in variance between cell types? While it is nice that genes identified are associated with pluripotency / differentiation, were there other characteristics to these genes? For example, were they particularly highly expressed or variable overall? This is a typical limitation of many single-cell analyses (only highly expressed genes can be studied), and it should be stated whether the method also has this limitation or can successfully rescue lowly expressed genes.

Line 11: "most studies ignore variation across individual donors" I am not sure this is a fair statement. More than one single-cell study highlight large intra-individual / intra-cell line variation (eg. Cuomo et al. ref. 15, Fig.1b, Eling et al PMID: 28360329). Additionally, other (admittedly simpler) methods exist that are similar to the proposed (e.g., muscat PMID: 33257685).

This is mentioned in the discussion already, but I find the choice of using pseudobulk quite limiting, as it means that single cells need to be somewhat arbitrarily discretized into separate groups that may not always reflect real biology. Do the authors have any plans of extending this to full single-cell profiles, e.g. by considering a Poisson GLMM?

Line 13. Typo: either "that underlie" or "underlying".

Reviewer #2 (Remarks to the Author):

Chen and Dahl introduce Cell Type-specific linear Mixed Model (CTMM). It is a method to model inter-individual variance in single cell expression data sets. The model allows for detecting (celltype specific) variance differences between individuals. The paper is well written and the methodology is clearly described. While this is potentially an interesting problem, the biological significance of these variance differences should be established more clearly in the analysis of real data sets. Can the authors provide evidence as to which biological phenomenon or mechanism could be underlying these differences? The novelty and merits of the method in comparison to other methods should be described and demonstrated more clearly.

Major comments:

1) One key motivation of the paper stated in the abstract is that "Modelling this inter-individual variation could improve statistical power to detect cell type-specific biology". So far the application to actual data is rather superficial and any new cell type-specific biology that might potentially be revealed is not described in any detail. It would strengthen the paper substantially, if the authors could demonstrate how to use their model to detect new cell type-specific biology. One way to do so would be to analyse a data set that contrasts patients and controls (a few of those have recently been published).

2) Linear mixed models have been proposed and applied the single cell field (see methods in Crowell et al. Nat Comm 2020 or Cuomo et al. Mol Syst Biol 2022). Crowell et al. proposed a simulation framework that can produce realistic multi-sample scRNA-seq data sets with different types of effects that they termed "differential states" (DS). In particular the differential proportion (DP), differential modality (DM) and the combination thereof (DB) would be relevant to the variance decomposition. Based on these simulations they compare different approaches. To fully judge the merits of CLMM it would be important to compare to these methods based on the same simulation framework (please also include cell regmap). How does the proposed model compare to these approaches?

3) The authors need to address more specifically the characteristics of single cell data. What is the effect of the number of cells and the number of reads sequenced for each cell on the accuracy and power of the approach?

a) By aggregating on the pseudo bulk level the authors make use of the central limit theorem to motivate the use of normal distributions. Please quantify what is the minimal number of cells needed, so that this approximation is reliable.

b) In addition, also the expression levels are quantified reliably only for the genes for highest expression levels. The other genes often have low counts and it is difficult to distinguish lowly expressed genes from non-expressed genes. What is the effect of mean expression (and number of reads per cell) on the power and accuracy of the approach?

c) please also demonstrate that the tool can be applied to data produced using the 10X genomics platform, as this is one of the most widely used at the moment and differs in terms of read depth from Smartseq

Minor:

4) The description of the overall pseudo bulk is quite extensive. Please motivate, why this model is useful in comparison to the much more powerful cell type specific pseudo bulk model.

5) In the introduction, when motivating the approach: make sure to make the distinction between sample level and cell level DE more clear.

6) How does the performance / accuracy of the model change for larger number of cell types?

7) The authors state that: "REML (JK), that is jackknife-based Wald test in REML, and HE were slightly inflated in CTP when sample size was 100 or lower." Sample size is likely to stay below 100 in many real life data sets. So what to do about it?

In the discussion the conclusion is that jack knife is the most robust. How does this fit with the observation of inflation?

8) Results of Figure 2:

- What is the "effect size" for these results: Its true positive rate reached above 80% even when sample size was only 20 and reached 100% when sample size was 50.
- What is the interpretation of the variances in figure S3 and S6? What is its scale? Would it make sense to express them relative to the overall variance?
- HE also had good power with about 50% of positive rate. Is 50% really a good power?
- Figure 2: please make the x-axis comparable, so that one can see the trade off between true positive and false positive rate for the same measure of variation ("effect size").
- Could you also create a figure with FDR and PPV plotted in the same plot.

9) Is there a way to estimate how large the noise of this parameter estimate v_{ic} is in reality? This would be quite important to understand in which conditions the method really works. Is there some kind of diagnostics that one could use on a data set to see if the model would be expected to work on that data set?

10) Would it make sense to optimize the v_{ic} in the ML model? The single cell estimate could serve as a starting value, or it could be used to define a prior on the parameter in a Bayesian setting

11) When doing the variance decomposition there is 14% variance between individuals within celltypes, 12.3% difference in mean expression (is this also cell type specific or individual specific) and 9% noise. Does this mean that the remaining variance is due to cell type composition? Could subcelltype composition explain part of the 14%?

12) Genes with high co-variance between celltypes and high inter-individual variance could be viewed as those that are differentiating different persons consistently across time. Are those genes known to be responsible for different developmental outcomes or do they share any other specific functional roles?

We thank the reviewers for their suggestions, which have significantly improved our paper. In addition to our point-by-point responses below, we highlight the three major additions we made in our revision:

- An analysis of a large, 10X-based dataset from Yazar et al 2022 Science (Figs S19-S29). This was a substantial effort.
- Simulations to characterize CTMM's scalability and dependence on scRNA-seq parameters (Figs 1 and S11)
- Much better references to prior scRNA-seq work, including a new Methods section and Table S1.

The first two changes address several major reviewer concerns around the utility of CTMM for future large-scale scRNA-seq datasets. The latter addresses several important minor requests to clarify why CTMM is fundamentally distinct from prior applications of LMMs to scRNA-seq data.

We use blue text to indicate changes that we made in the manuscript itself.

Reviewer #1 (Remarks to the Author):

Chen and Dahl propose CTMM, a linear mixed model framework to test for cell type-specific intra-individual variation in single-cell data from multiple individuals. Overall I think this model is well presented and robust, and a useful addition to the field.

My main concern is around performance in common droplet-based single-cell data, both in terms of the distribution of (pseudobulk) counts, and scalability. I also believe the author may have overstated the method's novelty a little.

We appreciate the positive view on CTMM's rigor and utility. We agree that it is important to profile CTMM more broadly, especially in terms of read counts and scalability, so we added an analysis of real droplet-based data (OneK1K) and several new simulations. While we are happy that the CTMM model itself was clear, we agree that we did not sufficiently describe CTMM's relationships to prior work, so we added a new Methods section and Supp Table. Overall, we appreciate the constructive criticisms, which substantially improved our work.

Main Comment

1. The authors use both simulated and real data to evaluate performance of the method. In the simulation experiments, synthetic data was generated from the models themselves, and a few parameter values were estimated. While I appreciate that not every single parameter value can be tested, and that some effort was put into making the simulations more realistic by estimating values from the iPSC data, I think there are a few aspects that should be taken into consideration. First, the choice of using 4 cell types seems arbitrary, and particularly low. I would argue most scRNA-seq studies identify 10+ clusters (see also my comment below on discrete cell types), so performance should be evaluated for a higher number of cell subpopulations (types). Second, the authors repeatedly mention the small sample sizes in single-cell data. While this is absolutely true compared to population genetic studies, there are now single-cell datasets from several hundreds of individuals – e.g., Yazar et al, ref. 18 includes nearly 1,000 individuals. How does CTMM perform on such sample sizes? The model should be tested on a larger dataset, or at least simulations should include sample sizes up to 1,000. Third, I may have missed it, but was the number of cells per individual varied when estimating pseudobulk? In real settings this varies widely and it would be good to evaluate the model in different circumstances. Also, why was a single (randomly drawn) gene used for more realistic simulations? What was its distribution? At least a few genes should be considered here, chosen at different overall mean and variance levels. Finally, and in relation to the real data used. Cuomo et al use plate-based scRNA-seq data

(SmartSeq2), which behave quite differently to droplet-based data, which is by far more popular today. In particular, this is full-length data and results in more reads per cells, but fewer cells in general (just over 36,000 cells, compared to well over a million in e.g., refs 16-18). I think performance on e.g., 10X scRNA-seq data (for example, refs. 16-18) is needed and would ensure the method will be more generalisable to real settings.

We agree with all of this. Thus, we added two independent (and substantial) analyses: (1) new simulations to address each dimension identified by the reviewer, and (2) a new real data analysis of droplet-based data.

New simulations varying # of cell types, sample size, # of cells, and read depth

1) Number of cell types, C:

We added a simulation that varies C from C=4 up to C=12 (**Supplementary Figure S7**, shown below). As desired, CTMM's primary test is calibrated for all C (panel A, 'REML (JK)') and its power increases with C (panel B). (While the LRT is inflated for larger C, we do not use this standard test in CTMM because it is inflated in practice, see Fig 2.) We summarize in the main text (lines 298-300):

As expected, the power increased when the main cell type became more common, when additional cell types were included, or when cell type-specific variance increased (**Supplementary Figure S7**).

2) Larger sample size, N:

We agree this is important, especially given the recent release of several large scRNA-seq datasets (Yazar 2022 Science, Perez 2022 Science, Nathan 2023 Nature). Thus, we increased the maximum N in our simulations from N=300 to N=1,000 (**Fig 1**, copied below). As expected, our REML (JK) test remains calibrated, and power increased with N.

Related minor writing change: We agree that we called scRNA-seq sample sizes ‘modest’ without sufficient context. Our point is only that standard asymptotics are not reliable, e.g., the traditional Wald test based on the likelihood’s Hessian is miscalibrated (blue lines in Fig 1A). We now use more careful language (lines 60-64):

We explored several statistical methods to fit and test CTMM’s parameters. Achieving calibrated and unbiased estimates in CTMM is challenging because scRNA-seq datasets currently have small to moderate numbers of donors, ranging from one to hundreds, and thus off-the-shelf asymptotic tests may fail.

3) # of cells and 4) read depth:
(Please note that the below is copied in response to R2’s major point 3 below.)

We added a new set of simulations operating at the level of single cells to evaluate CTMM as a function of #cells and read depth. (Our primary simulations operate at the pseudobulk level for computational efficiency). Our simulation directly uses the Cuomo 2020 data because realistically simulating scRNA-seq remains challenging (see, e.g., Crowell et al 2023 Genome Biology).

We describe this simulation fully in the new **Supplementary Note Section 3.4, “Simulation of single-cell gene expression.”** The outline of the simulation for CTMM’s “Hom” model is:

1. To ensure the Hom model holds, subset the Cuomo 2020 data to cells from a single cell type
2. Simulate reads for gene g in cell s from a Binomial(R_s , p_{gs}) distribution, where
 - R_s = total number of reads in cell s
 - p_{gs} = proportion of reads in cell s from gene g
 This simulates a cell matching cell s in terms of expression distribution (p_{gs}) and read depth (R_s)
3. For each individual, draw cells randomly from the individual’s pool of simulated cells – this enables simulating #cells greater than is observed in reality
4. Aggregate the single cell data into pseudobulk data (i.e., construct Y and ν) and input this into CTMM

We evaluate these Hom simulations as in Figure 2A, where we vary the level of error in our estimates of ν . First, we vary read depth to assess much lower read depths (down to .01x the real data) because the Cuomo 2020 and Yazar 2022 data have very different read depths (~.5M/cell vs ~3K/cell, respectively). CTMM is calibrated across this full range of read depths (REML (JK), **Supp Fig 11A**, copied below). Second, we vary the number of cells from .5x to 2x the real data (Cuomo 2020 and Yazar 2022 are similar—72 vs 92 cells per cell type per individual). Overall, CTMM is generally robust to the number of cells (**Supp Fig 11C**). While

CTMM is slightly inflated for .5x cells when ν noise is low, this is not a practically significant problem because modern droplet-based data usually have more cells per individual, not fewer (e.g., Yazar 2022 has >4x more than Cuomo 2020). Additionally, CTMM is calibrated for realistic levels of noise in ν , especially given that the reference quantiles of $CV(\nu)$ (vertical lines in Supp Fig 11A,C) are conservatively estimated in the real data with 1x cells. Overall, we conclude that CTMM is robust for droplet-based data.

To simulate from CTMM's Free model, where interindividual variance is cell type-specific, we randomly permute each column of Y (and apply the same permutations to the analogous columns of ν). As expected, CTMM power increases with read depth (**Supp Fig 11B**) and the number of cells (**Supp Fig 11D**).

In addition to the full description in Supp Note 3.4, we summarize these results in the Main text (lines 338-341):

Finally, we conducted simulations at the level of single cells to evaluate the impact of sequencing depth and the number of cells (Supplementary Note Section 3.4). We found that CTMM was robust across a realistic range for sequencing depth and number of cells (Supplementary Figure S11 A, C), though power improved with greater read depth or number of cells (Supplementary Figure S11 B, D).

Related point: “Why just one gene?”

Apologies, our writing was unclear. We agree this would be wrong. But, actually, we pick 1,000 genes at the outset and then perform one simulation replicate per each of these genes (per parameter setting). We have now fixed the relevant sentence (in Supp Note Section 3.3):

For each simulation setting, we ran 1,000 replicates where each uses parameters estimated from a distinct gene in a predefined set of 1,000 randomly chosen genes.

In the simulation of the Hom model, for each gene, we chose its parameters (σ_{α}^2 and β) to match the real CTMM estimates from the iPSCs data (estimated using REML under the Free model).

Related point: Droplet vs plate-based

While the new simulations we added above partly address this concern by varying N, read depth, and cell number—which differ between these technologies—we recognize that simulations are no substitute for real data. Therefore, we took your suggestion and analyzed the OneK1K data.

New analysis of real droplet-based data

We complemented our new simulations with an analysis of the OneK1K data from Yazar et al 2022 Science. This dataset is representative of next-generation scRNA-seq datasets, e.g. the sequencing is droplet-based (10X) and, depending on QC choices, N is ~500 to ~1,000 and C is ~6 to ~14. Our analysis is fully described in **Supplementary Note Section 4 (‘PBMC analysis’)**. The transcriptome-wide CTMM estimates and tests are shown in **Figure S22** and **Figure S20** (both copied below). We describe our results in a new Results section (lines 452-489, copied below). **Overall, we find that CTMM can be applied to recent 10X-based data.**

Application to peripheral blood mononuclear cells

The iPSC data¹³ we analyzed above used plate-based sequencing. We next sought to confirm that CTMM is applicable to droplet-based sequencing, a different scRNA-seq technology that generally trades off a greater number of cells at the cost of lower read depth. We thus applied CTMM to the recent droplet-based data from the OneK1K cohort²². This dataset has many more individuals than the iPSC data (N=982 vs 125) and more cells per individual (1,300 vs 300), but it has far fewer reads per cell (~3K vs ~0.5M). The cells themselves are also very different, as OneK1K contains peripheral blood mononuclear cells (PBMCs) from living people rather than differentiating iPSCs from a controlled lab experiment. Another important difference is that the PBMC cell types are computationally inferred, while the iPSC cell types are defined by experimental days. Finally, the PBMC cell type proportions vary substantially (Supplementary Figure S19). Our primary analysis restricted to cell types with at least 10 cells in at least 90% of individuals, resulting in 7 cell types (CD4_{NC}, CD4_{ET}, CD8_{ET}, CD8_{NC}, NK, B_{IN}, and B_{Mem}, Supplementary Note Section 4).

We find that CTMM provides powerful and robust estimates in the OneK1K data. First, CTMM detected significant cell type-specific interindividual variance for 2,310 genes out of the 11,526 total genes tested ($p < .05/11526$, Supplementary Figure S20). The top signal is RPS26 ($p = 1.78 \times 10^{-167}$, Supplementary Figure S21), which plays a key role in regulating T cells^{47,48}. Further, genetic variation causes interindividual variation in this gene that is cell type-specific²² and is linked to complex traits such as eczema and asthma⁴⁹. Second, CTMM partitions transcriptome-wide interindividual variation into components that are shared across cell types (10.9%) vs cell type-specific (21.5% on average across cell types, Supplementary Figure S22). This is an interesting contrast with the iPSC results, where shared interindividual variation was near zero. Biologically, this shows that the shared interindividual context varies far more for PBMCs than for iPSCs, which is plausible because the former

are from living people while the latter are from a controlled experiment. Finally, we evaluated CTMM's Full model transcriptome-wide to quantify interindividual covariance between cell types. These estimates recapitulated expected relationships between cell types (Supplementary Figure S23). For example, the most-correlated cell types are CD4_{NC} and CD4_{ET}; intuitively, this means that an individual with above-average expression of a gene in their CD4_{NC} cells will typically also have above-average expression in their CD4_{ET} cells. The second most-correlated cell types are CD4_{NC} and CD8_{NC}, which is consistent with the observation that CD4_{NC} and CD8_{NC} shared the most genetic effects in prior work²². We next tested the robustness of CTMM to rarer cell types in a secondary analysis that includes two additional cell types, Mono_C and CD8_{S100B}. CTMM gave consistent results for the 7 larger cell types that are included in both analyses (Supplementary Figure S24-S28). As expected, CTMM's estimates are noisier for Mono_C and CD8_{S100B}, which are rarer cell types. Nonetheless, adding these cell types enables CTMM to discover new differentially-variable genes. For example, the top newly-significant gene is TMEM176B ($p = 5.45 \times 10^{-63}$ vs $p=0.18$ in our primary analysis), which makes sense as this gene is primarily expressed in Mono_C (Supplementary Figure S29). We conclude that CTMM's results are robust to variations in the input cell types, but its estimates are less accurate for rarer cell types.

Figure S20. Transcriptome-wide distribution of CTMM p values in OneK1K using HE. CTMM's test for cell type-specific interindividual variation is on the y-axis; its test for mean differential expression across cell types is on the x-axis. Each dot represents a gene, and colors reflect the density of genes in the area, with yellow indicating higher density. Dashed lines indicate the Bonferroni significance

thresholds. Top right insert: the distribution of p-values from CTMM's interindividual variance test is shown for genes where the p-value for mean differential expression is essentially 0.

Figure S22. Transcriptome-wide distribution of CTMM's estimates for cell type-specific interindividual variance in OneK1K. The homogeneous variance shared across cell types (σ_α^2) and the cell type-specific variance (V_{CT}) are estimated from the Free model using HE. Violins were truncated to (-2, 2) for visibility. Cell types are ordered by cell type proportion. Interior box plots show the median and the first and third quartiles. The whiskers extend to values within 1.5 times the interquartile range from the first and third quartiles.

Other Comments

- Can the authors comment on the genes that were found to be significantly differentiated in variance between cell types? While it is nice that genes identified are associated with pluripotency / differentiation, were there other characteristics to these genes? For example, were they particularly highly expressed or variable overall? This is a typical limitation of many single-cell analyses (only highly expressed genes can be studied), and it should be stated whether the method also has this limitation or can successfully rescue lowly expressed genes.

To answer this, we compared the top 1,000 genes vs the bottom 1,000 genes based on CTMM p-values and compared them in terms of (A) overall expression, (B) cell type-specific expression, (C) overall variance, and (D) differential mean expression across cell types. These results are in Figure S30 (copied below)

- A. Overall mean expression: As the reviewer predicted, genes with greater significance in CTMM have higher expression.
- B. Cell type-specific mean expression: We found qualitatively identical results to the overall expression analysis in A: Mean expression affects power, but there is no particularly dominant cell type.
- C. Overall expression variance: We again found higher levels in the top CTMM genes. This effect is highly statistically significant, as expected—power to distinguish variance components, overall variance, and total expression should scale together. The effect is notably weaker than that of mean expression in A.
- D. Differential mean expression: We found that the differential-variance genes from CTMM are enriched in differential-mean genes. This is expected: Biologically, genes differing between cell types are likely to differ in complex ways that affect both mean and variance (and even higher-order moments); Statistically, both tests' power is affected by gene-level features (especially overall expression level).

These analyses improve our characterization of CTMM's performance, and we summarize them in the Discussion. We focus on the role of overall expression, as in the reviewer's comment (lines 521-523):

We find that higher overall levels of a gene's expression increase the power of CTMM (Supplementary Figure S30), which also holds for most tests of scRNA-seq data.

3. Line 11: “most studies ignore variation across individual donors” I am not sure this is a fair statement. More than one single-cell study highlight large intra-individual / intra-cell line variation (eg. Cuomo et al. ref. 15, Fig.1b, Eling et al PMID: 28360329). Additionally, other (admittedly simpler) methods exist that are similar to the proposed (e.g., muscat PMID: 33257685).

We completely agree. We respond in two parts: first, fixing important errors in our writing; second, explaining in detail the relationships between CTMM and prior LMMs applied to scRNA-seq. (The latter is more relevant to Reviewer 2’s comments, but we include it here for completeness.)

First, we used overly broad language. We specifically meant variance component models, but “variation” has much broader implications than we intended (“variation” even includes ordinary differential mean expression tests, which are very widely studied). We have corrected this sentence in Abstract (line 11):

... most studies do not consider cell type-specific variation across donors.

We think this appropriately emphasizes that CTMM’s novelty is its focus on cell type-specificity. This also fixes our incorrect (and accidental) implication that CTMM is the first LMM applied to scRNA-seq.

We further clarify this claim in the Introduction, where we have more space (lines 38-41):

Several studies have applied linear mixed models to scRNA-seq data to account for variance across individuals, cell types, or experimental batches¹⁰⁻¹⁴, but this variance has not been unbiasedly partitioned across cell types, and the potential for bias and miscalibration have not been evaluated.

Again, this emphasizes that the novelty of our work is the particular LMM that we use to partition interindividual variation across cell types into shared vs specific components, while prior LMMs applied to scRNA-seq use fundamentally different models. To be clear, this is **not** a criticism of these prior studies—they are neither better nor worse than CTMM, they just ask different questions and thus are not comparable.

We also made a related writing change in the Discussion (CTMM opens the door -> CTMM is a step toward).

Second, we copy below our response to R2’s Major Point 2 on clarifying CTMM’s relationship to prior work:

We fully agree that we under-referenced prior literature, especially given that LMMs are a basic model in statistics and that they have been previously applied to scRNA-seq data. We have addressed this by adding (1) a new Supplementary Table describing key features of CTMM relative to prior LMMs for scRNA-seq and (2) a new section in the Methods to summarize these relationships.

Table S1 (copied below) includes all of the references from both reviewers (and also Tung et al 2017). The point of CTMM is to decompose interindividual variation across cell types into shared and specific components, which prior studies cannot do (**column 3**): they either fit each cell type separately, ignoring shared variation across cell types (Martinez-Jimenez 2017), or they assume that all cell types have a predefined covariance (Tung 2017, Cuomo 2020, and Crowell 2020 assume it is identical across cell types, akin to our Hom model, while Cuomo 2022 fixes it to a transcriptome-wide average matrix). On a technical level, CTMM derives and validates novel tests for cell type-specificity (**column 4**), and also uses entirely novel inference methods to estimate variance components (**column 5**). A final novelty is that CTMM uses pseudobulk (**column 2**), which is conceptually straightforward but practically crucial—cell-level models cannot scale to either of the real datasets that we use. We summarize this in a new Methods section (lines 186-196):

Prior LMM applications to scRNA-seq data

Linear mixed models (LMMs) are a basic statistical framework for partitioning variation, and several prior studies have applied LMMs to scRNA-seq data (**Table S1**). Most prior work has applied generic LMM methods at the level of single cells. They fit variance components for batch effects^{11,13}, experimental context¹⁰ and/or some form of interindividual variation¹¹⁻¹⁴. Additionally, some of these studies model the non-Gaussianity of cell-level expression data^{10,14}. Despite the strengths of these works, none aim to partition shared vs specific components of interindividual variation. This is the key novelty in CTMM, and it requires a different variance component model than the ones that have been used in prior work. At a more technical level, CTMM develops a novel testing framework based on jackknife rather than use off-the-shelf tools, which solves biases in standard LMM variance component tests due to the complexity of scRNA-seq data.

Paper	Pseudobulk or cell level	Cross-cell type variance model	Variance component test	Inference method
Tung 2017 ¹	Cell	Hom*	No	blmer ²
Martinez-Jimenez 2017 ³	Cell	Pseudo-Free**	Yes***	BASiCS ⁴
Crowell 2020 ⁵	Cell	Hom	No [^]	bglmer ² /lmer ⁶ / variancePartition ⁷
Cuomo 2020 ⁸	Cell	Hom	No	LIMIX ⁹
Cuomo 2022 ¹⁰	Cell	Pseudo-Full**	No ^{^^}	CellRegMap ¹⁰
CTMM	Pseudobulk	Free + Full	New jackknife test	New methods to fit REML and method-of-moments

*In the special case where cell types are experimental batches and each batch contains exactly one donor, the batch effect model is similar to the simplified Free model where all cell types have equal variance (Supplementary Figure S31). **We use “Pseudo” for models that cannot decompose interindividual variation: “Pseudo-Free” omits the Hom component because cell types are each studied separately; “Pseudo-Full” assumes a pre-defined covariance across cell types that is identical for all genes. ***Applies to N=1 individual at a time. [^]The “differential state” test does not distinguish cell type-specific expression in mean vs variance (x- vs y-axes in Figure 4). ^{^^}CellRegMap is a method to estimate and test eQTLs and does not apply to the variance components in CTMM.

We now provide a brief summary for each of these references below to justify our broad summary conclusions. These are not criticisms—these papers just decompose variance in different ways than CTMM.

Martinez-Jimenez, Eling et al 2017 Science:

This paper shows the importance of variance in a biomedically significant model system and is thus an excellent reference for the core concept in CTMM. This paper does not use LMMs, but rather decomposes variance by applying a sophisticated variance model (BASiCS) to each experimental condition separately and then testing their differences post-hoc. While this is fully rigorous, it does not model variation across individuals (see BASiCS equation 1 in Vallejos et al Genome Biology 2016, which indexes by cell, gene, and cell type—but not by individual). Additionally, by only studying one cell type (i.e., condition) at a time, it cannot model shared variation across cell types. Neither is a problem in Martinez-Jimenez et al: they focus on comparing two conditions (young vs old) in inbred strains, so N=1 is sufficient, and they focus on differences in total variance, which does not require an unbiased partition of variance.

Tung et al 2017 Sci. Rep.

This work does investigate variance components in scRNA-seq data. However, the model restricts to nonnegative variance components, which severely biases transcriptome-wide average estimates (Steinsaltz et al. Genetics 2020). Additionally, the model treats single cell-level data as Gaussian, which is inappropriate (cf R1 point 4). Most importantly, the model does not partition cell type-level variation. Its model for batch effects does resemble our simplified Free model in interesting ways, but this only holds for very non-standard experimental designs. For example, it holds in this paper because the experiment was designed to study batch effects, hence each individual was studied in multiple technical replicates. This model cannot be applied to real cell types because they are not technical replicates. We are currently working with the Gilad lab to incorporate CTMM in their current analyses.

Crowell 2020 NC:

This paper develops a software package (*muscat*) that implements many useful functions to simulate and analyze scRNA-seq data. It implements standard (G)LMMs with a single variance component for inter-individual variation and applies this to each cell type one-at-a-time. (No explicit model is given, but the methods include lmer pseudo-code with (1|id), which we believe is an individual-level effect that does not depend on cell type.) As with Tung 2017 and Cuomo 2020, this cannot capture cell type-specific variation. Additionally, we do not believe this paper ever tests differential variance; rather, it tests for “differential state” which is a conglomerate of differences in mean, variance, and (sub-)cell type proportion—this is valid but does not demonstrate differential variance. We also believe it always tests fixed-effect mean differences, and never evaluates the random effects. Overall, the unambiguous difference is CTMM partitions shared vs specific variation across cell types, but the *muscat* LMMs do not.

Cuomo 2020 NC:

This paper is the source of our primary data. It applies a standard Gaussian LMM to cell level data, as implemented in LIMIX. This is very similar to the decomposition in Tung 2017: it fits only shared interindividual variance; it inappropriately assumes Gaussianity; it is not scalable (in fact, they must drastically downsample their data to fit LIMIX: “To reduce computational cost, we considered a random subset of 5000 cells”); and it is systematically biased by requiring nonnegative variance components. (This paper also studies eQTLs, but this is very different from CTMM.) We do not consider these to be important problems as that paper focused on the real data, and we view its LMM analysis just as a helpful illustration of the data (one subpanel of one main Figure).

Cuomo 2022 MSB:

This paper drastically scales up eQTL analyses from Cuomo 2020 using a computational trick from the genetics LMM literature (from StructLMM, Moore et al Nat Gen 2018). This is an impactful, rigorous and relevant reference. But the method itself, CellRegMap, is not comparable to CTMM. CTMM does not consider genetics, and CellRegMap does not estimate or test interindividual variance components. Nonetheless, CellRegMap does include an interindividual variance model with similarities to CTMM, which we previously missed. Specifically, the “u” term in their first Methods equation is an interindividual variance component. Nonetheless, this term assumes that interindividual variance components are already known (in practice, Cuomo 2022 assumes it is (essentially) just the PCs of transcriptome-wide expression). This means CellRegMap cannot partition interindividual variation; one way to see this is that CellRegMap assumes every gene has identical covariance across cell types. While we assume this is not problematic for CellRegMap’s eQTL test, it does show that CellRegMap is not comparable to CTMM. (And CellRegMap does not provide tests for non-eQTL variance components for us to assess.)

Overall, we now understand that CellRegMap is highly relevant, but it is not comparable to CTMM.

4. This is mentioned in the discussion already, but I find the choice of using pseudobulk quite limiting, as it means that single cells need to be somewhat arbitrarily discretized into separate groups that may not always reflect real biology. Do the authors have any plans of extending this to full single-cell profiles, e.g. by considering a Poisson GLMM?

We agree—pseudobulk inherently throws away data, sacrificing statistical power and cell-level resolution. We also agree this is worth being more explicit about, so we now say in the Discussion (lines 534-538):

Fifth, CTMM only considers pseudobulk data, which greatly improves computational efficiency and facilitates its simplifying Gaussian assumptions. Nonetheless, pseudobulk inherently discards cell-level information, which sacrifices statistical power and resolution within cell types. Moreover, cell type-specific pseudobulk necessarily discretizes cells into categories, which can be somewhat arbitrary.

We also agree GLMMs are the right solution in terms of math, stats, and biology—but they are very computationally challenging. In fact, even regular LMMs are challenging, e.g., Cuomo 2020 had to drastically downsample their data to fit their cell-level LMM. Instead, we plan to extend the pseudobulk model, e.g., genetic vs nongenetic, cross-gene correlations and networks, and cases vs controls. But if computationally tractable cell-level GLMM methods become available, we will port CTMM's pseudobulk models to the cell level.

Line 13. Typo: either “that underlie” or “underlying”.

Thank you, we have fixed this.

Reviewer #2 (Remarks to the Author):

Chen and Dahl introduce Cell Type-specific linear Mixed Model (CTMM). It is a method to model inter-individual variance in single cell expression data sets. The model allows for detecting (celltype specific) variance differences between individuals. The paper is well written and the methodology is clearly described. While this is potentially an interesting problem, the biological significance of these variance differences should be established more clearly in the analysis of real data sets. Can the authors provide evidence as to which biological phenomenon or mechanism could be underlying these differences? The novelty and merits of the method in comparison to other methods should be described and demonstrated more clearly.

We appreciate the reviewer's positive view on our writing and methodology. We also agree with their suggestions to apply CTMM to recent scRNA-seq datasets and to better reference prior applications of off-the-shelf LMMs to scRNA-seq data. These changes have significantly improved our manuscript.

To clearly state CTMM's novelty: CTMM partitions interindividual variance into shared vs specific components across cell types. We do not believe any prior LMM has done this. (There are also important technical novelties in CTMM.) But we fully agree that we did a poor job describing this important point, so we added a Supp Table and section in the Methods. We emphasize we are not criticizing prior LMMs applications to scRNA-seq data; CTMM takes LMMs in an orthogonal direction, hence it is complementary to these prior studies.

As to biological significance, we only claim to show proof-of-principle that CTMM can succeed in real data. Specifically, CTMM finds genes missed by ordinary differential expression tests (Fig 4) and these genes are enriched in plausible biological functions (Table 2). Also, CTMM shows that almost all interindividual variation across iPSC differentiation is cell type-specific (Fig 3). Further, our new analysis of OneK1K data shows CTMM can achieve similar success in 10X data (Supp Figs 19-29). We do not claim to discover a new biological phenomenon or mechanism; we only claim that CTMM informs a novel dimension of scRNA-seq data that can be biologically meaningful.

Major comments:

1) One key motivation of the paper stated in the abstract is that "Modelling this inter-individual variation could improve statistical power to detect cell type-specific biology". So far the application to actual data is rather superficial and any new cell type-specific biology that might potentially be revealed is not described in any detail. It would strengthen the paper substantially, if the authors could demonstrate how to use their model to detect new cell type-specific biology. One way to do so would be to analyse a data set that contrasts patients and controls (a few of those have recently been published).

We agree that our novel method is our primary contribution, not our novel biological results. We only aim to show proof-of-principle, which we think we achieve: CTMM finds novel cell type-specific results with plausible biology (Table 2), and CTMM partitions shared vs specific components of variation across cell types (Figure 3). Neither result is possible using prior tools, hence we consider them new forms of cell type-specific biology.

Further, we added an analysis of a recent whole blood dataset (OneK1K). This found additional signals of cell type-specific biology, including (1) two plausible genes with cell type-specific variation and links to complex disorders (*RPS26* and *TMEM176B*, highlighted in the Main text and Supp Figs S21 and S29) and (2) transcriptome-wide partitions of shared vs specific (co-)variation across cell types (Supp Figs S22 and S23). Comparing the latter to the Cuomo 2020 data reveals biological differences between whole blood vs iPSC cell types.

As a minor point, CTMM does not directly apply to case/control data: (1) CTMM assumes individuals are i.i.d, and (2) case/control data have important complexities, such as over-ascertainment of cases. We considered meta-analyzing CTMM results *post hoc* in cases vs controls, but this would require extensive robustness analyses warranting a separate paper. Nonetheless, we agree this is a great future direction, and we added it to the Discussion (lines 555-558):

It would also be useful to extend CTMM to test for variance differences between groups of individuals, e.g., disease cases and controls. Nonetheless, this will require careful modelling and robustness tests to account for subtle ascertainment biases.

2) Linear mixed models have been proposed and applied the single cell field (see methods in Crowell et al. Nat Comm 2020 or Cuomo et al. Mol Syst Biol 2022). Crowell et al. proposed a simulation framework that can produce realistic multi-sample scRNA-seq data sets with different types of effects that they termed "differential states" (DS). In particular the differential proportion (DP), differential modality (DM) and the combination thereof (DB) would be relevant to the variance decomposition. Based on these simulations they compare different approaches. To fully judge the merits of CLMM it would be important to compare to these methods based on the same simulation framework (please also include cell regmap). How does the proposed model compare to these approaches?

We completely agree that we under-referenced prior applications of off-the-shelf LMM tools to scRNA-seq data. Reviewer 1 also noted this, and we copy our shared response to this general point below. But first, we address specific points:

- Compare to "differential state": this is not possible. "Differential state" does not distinguish differences in mean vs variance. We cannot easily derive a new test from their output *post hoc*, either: this paper only applies (G)LMMs to the Hom model, hence it cannot learn cell type-specific variation.
- Compare to "CellRegMap": this is not possible. It is a model for cell type-specific eQTLs. While random effects are present, they function only to improve the eQTL test ("We note that the introduction of this additional covariance term...is critical for retaining calibration [of the eQTL test].") Concretely, CellRegMap does not test interindividual random effects nor does it learn covariance across cell types.
 - Technical answer in CellRegMap notation: It assumes the cross-cell type covariance Σ is known and learns a scalar weight on it (σ^2_{RC}); CTMM's goal is to unbiasedly estimate a general Σ (akin to our "V"). CellRegMap's model suffices for eQTL tests but not our tests.
- Re: *muscat* simulations: we instead performed simulations based directly on the real data from Cuomo 2020 (see Supp Fig S11), as we agree with subsequent work by the same group that simulations based on real data are superior (Crowell 2023 GenBio). Also, on a technical note, we do not think it is possible to simulate CTMM's model from *muscat*, as it does not distinguish differential mean vs variance.

(To be clear, these are not criticisms of *muscat* or CellRegMap, which neither claim to distinguish (1) differential mean vs variance nor (2) shared vs specific variation. "Differential state" and cell type-specific eQTLs are valid and important uses of (G)LMMs, they are just not comparable to CTMM.)

Nonetheless, we do agree (1) it is important to consider more realistic simulations and (2) it is important to better reference prior work. For the former, please see our response to your Major point 3 below. For the latter, we copy the shared response to this point and the similar point from Reviewer 1:

We fully agree that we under-referenced prior literature, especially given that LMMs are a basic model in statistics and that they have been previously applied to scRNA-seq data. We have addressed this by adding (1) a new Supplementary Table describing key features of CTMM relative to prior LMMs for scRNA-seq and (2) a new section in the Methods to summarize these relationships.

Table S1 (copied below) includes all of the references from both reviewers (and also Tung et al 2017). The point of CTMM is to decompose interindividual variation across cell types into shared and specific components, which prior studies cannot do (**column 3**): they either fit each cell type separately, ignoring shared variation across cell types (Martinez-Jimenez 2017), or they assume that all cell types have a predefined covariance (Tung 2017, Cuomo 2020, and Crowell 2020 assume it is identical across cell types, akin to our Hom model, while Cuomo 2022 fixes it to a transcriptome-wide average matrix). On a technical level, CTMM derives and validates novel tests for cell type-specificity (**column 4**), and also uses entirely novel inference methods to estimate variance components (**column 5**). A final novelty is that CTMM uses pseudobulk (**column 2**), which is conceptually straightforward but practically crucial—cell-level models cannot scale to either of the real datasets that we use. We summarize this in a new Methods section (lines 186-196):

Prior LMM applications to scRNA-seq data

Linear mixed models (LMMs) are a basic statistical framework for partitioning variation, and several prior studies have applied LMMs to scRNA-seq data (**Table S1**). Most prior work has applied generic LMM methods at the level of single cells. They fit variance components for batch effects^{11,13}, experimental context¹⁰ and/or some form of interindividual variation¹¹⁻¹⁴. Additionally, some of these studies model the non-Gaussianity of cell-level expression data^{10,14}. Despite the strengths of these works, none aim to partition shared vs specific components of interindividual variation. This is the key novelty in CTMM, and it requires a different variance component model than the ones that have been used in prior work. At a more technical level, CTMM develops a novel testing framework based on jackknife rather than use off-the-shelf tools, which solves biases in standard LMM variance component tests due to the complexity of scRNA-seq data.

Paper	Pseudobulk or cell level	Cross-cell type variance model	Variance component test	Inference method
Tung 2017 ¹	Cell	Hom*	No	blmer ²
Martinez-Jimenez 2017 ³	Cell	Pseudo-Free**	Yes***	BASiCS ⁴
Crowell 2020 ⁵	Cell	Hom	No [^]	bglmer ² /lmer ⁶ / variancePartition ⁷
Cuomo 2020 ⁸	Cell	Hom	No	LIMIX ⁹
Cuomo 2022 ¹⁰	Cell	Pseudo-Full**	No ^{^^}	CellRegMap ¹⁰
CTMM	Pseudobulk	Free + Full	New jackknife test	New methods to fit REML and method-of-moments

*In the special case where cell types are experimental batches and each batch contains exactly one donor, the batch effect model is similar to the simplified Free model where all cell types have equal variance (Supplementary Figure S31). **We use “Pseudo” for models that cannot decompose interindividual variation: “Pseudo-Free” omits the Hom component because cell types are each studied separately; “Pseudo-Full” assumes a pre-defined covariance across cell types that is identical for all genes. ***Applies to N=1 individual at a time. [^]The “differential state” test does not distinguish cell type-specific expression in mean vs variance (x- vs y-axes in Figure 4). ^{^^}CellRegMap is a method to estimate and test eQTLs and does not apply to the variance components in CTMM.

We now provide a brief summary for each of these references below to justify our broad summary conclusions. These are not criticisms—these papers just decompose variance in different ways than CTMM.

Martinez-Jimenez, Eling et al 2017 Science:

This paper shows the importance of variance in a biomedically significant model system and is thus an excellent reference for the core concept in CTMM. This paper does not use LMMs, but rather decomposes variance by applying a sophisticated variance model (BASiCS) to each experimental condition separately and then testing their differences post-hoc. While this is fully rigorous, it does not model variation across individuals (see BASiCS equation 1 in Vallejos et al Genome Biology 2016, which indexes by cell, gene, and cell type—but not by individual). Additionally, by only studying one cell type (i.e., condition) at a time, it cannot model shared variation across cell types. Neither is a problem in Martinez-Jimenez et al: they focus on comparing two conditions (young vs old) in inbred strains, so $N=1$ is sufficient, and they focus on differences in total variance, which does not require an unbiased partition of variance.

Tung et al 2017 Sci. Rep.

This work does investigate variance components in scRNA-seq data. However, the model restricts to nonnegative variance components, which severely biases transcriptome-wide average estimates (Steinsaltz et al. Genetics 2020). Additionally, the model treats single cell-level data as Gaussian, which is inappropriate (cf R1 point 4). Most importantly, the model does not partition cell type-level variation. Its model for batch effects does resemble our simplified Free model in interesting ways, but this only holds for very non-standard experimental designs. For example, it holds in this paper because the experiment was designed to study batch effects, hence each individual was studied in multiple technical replicates. This model cannot be applied to real cell types because they are not technical replicates. We are currently working with the Gilad lab to incorporate CTMM in their current analyses.

Crowell 2020 NC:

This paper develops a software package (*muscat*) that implements many useful functions to simulate and analyze scRNA-seq data. It implements standard (G)LMMs with a single variance component for inter-individual variation and applies this to each cell type one-at-a-time. (No explicit model is given, but the methods include lmer pseudo-code with (1|id), which we believe is an individual-level effect that does not depend on cell type.) As with Tung 2017 and Cuomo 2020, this cannot capture cell type-specific variation. Additionally, we do not believe this paper ever tests differential variance; rather, it tests for “differential state” which is a conglomerate of differences in mean, variance, and (sub-)cell type proportion—this is valid but does not demonstrate differential variance. We also believe it always tests fixed-effect mean differences, and never evaluates the random effects. Overall, the unambiguous difference is CTMM partitions shared vs specific variation across cell types, but the *muscat* LMMs do not.

Cuomo 2020 NC:

This paper is the source of our primary data. It applies a standard Gaussian LMM to cell level data, as implemented in LIMIX. This is very similar to the decomposition in Tung 2017: it fits only shared interindividual variance; it inappropriately assumes Gaussianity; it is not scalable (in fact, they must drastically downsample their data to fit LIMIX: “To reduce computational cost, we considered a random subset of 5000 cells”); and it is systematically biased by requiring nonnegative variance components. (This paper also studies eQTLs, but this is very different from CTMM.) We do not consider these to be important problems as that paper focused on the real data, and we view its LMM analysis just as a helpful illustration of the data (one subpanel of one main Figure).

Cuomo 2022 MSB:

This paper drastically scales up eQTL analyses from Cuomo 2020 using a computational trick from the genetics LMM literature (from StructLMM, Moore et al Nat Gen 2018). This is an impactful, rigorous and relevant reference. But the method itself, CellRegMap, is not comparable to CTMM. CTMM does not consider genetics, and CellRegMap does not estimate or test interindividual variance components. Nonetheless, CellRegMap does include an interindividual variance model with similarities to CTMM, which we previously missed. Specifically, the “u” term in their first Methods equation is an interindividual variance component. Nonetheless, this term assumes that interindividual variance components are already known (in practice, Cuomo 2022 assumes it is (essentially) just the PCs of transcriptome-wide expression). This means CellRegMap cannot partition interindividual variation; one way to see this is that CellRegMap assumes every gene has identical covariance across cell types. While we assume this is not problematic for CellRegMap’s eQTL test, it does show that CellRegMap is not comparable to CTMM. (And CellRegMap does not provide tests for non-eQTL variance components for us to assess.)

Overall, we now understand that CellRegMap is highly relevant, but it is not comparable to CTMM.

3) The authors need to address more specifically the characteristics of single cell data. What is the effect of the number of cells and the number of reads sequenced for each cell on the accuracy and power of the approach?

We fully agree, and this is very similar to parts of R1’s main concern. Therefore, we added a new set of simulations to evaluate the number of cells and read depth. We copy that text here:

We added a new set of simulations operating at the level of single cells to evaluate CTMM as a function of #cells and read depth. (Our primary simulations operate at the pseudobulk level for computational efficiency). Our simulation directly uses the Cuomo 2020 data because realistically simulating scRNA-seq remains challenging (see, e.g., Crowell et al 2023 Genome Biology).

We describe this simulation fully in the new **Supplementary Note Section 3.4, “Simulation of single-cell gene expression.”** The outline of the simulation for CTMM’s “Hom” model is:

5. To ensure the Hom model holds, subset the Cuomo 2020 data to cells from a single cell type
6. Simulate reads for gene g in cell s from a Binomial(R_s, p_{gs}) distribution, where
 - R_s = total number of reads in cell s
 - p_{gs} = proportion of reads in cell s from gene gThis simulates a cell matching cell s in terms of expression distribution (p_{gs}) and read depth (R_s)
7. For each individual, draw cells randomly from the individual’s pool of simulated cells – this enables simulating #cells greater than is observed in reality
8. Aggregate the single cell data into pseudobulk data (i.e., construct Y and ν) and input this into CTMM

We evaluate these Hom simulations as in Figure 2A, where we vary the level of error in our estimates of ν . First, we vary read depth to assess much lower read depths (down to .01x the real data) because the Cuomo 2020 and Yazar 2022 data have very different read depths (~.5M/cell vs ~3K/cell, respectively). CTMM is calibrated across this full range of read depths (REML (JK), **Supp Fig 11A**, copied below). Second, we vary the number of cells from .5x to 2x the real data (Cuomo 2020 and Yazar 2022 are similar—72 vs 92 cells per cell type per individual). Overall, CTMM is generally robust to the number of cells (**Supp Fig 11C**). While CTMM is slightly inflated for .5x cells when ν noise is low, this is not a practically significant problem because modern droplet-based data usually have more cells per individual, not fewer (e.g., Yazar 2022 has >4x more than Cuomo 2020). Additionally, CTMM is calibrated for realistic levels of noise in ν , especially given that the

reference quantiles of $CV(\nu)$ (vertical lines in Supp Fig 11A,C) are conservatively estimated in the real data with 1x cells. Overall, we conclude that CTMM is robust for droplet-based data.

To simulate from CTMM's Free model, where interindividual variance is cell type-specific, we randomly permute each column of Y (and apply the same permutations to the analogous columns of ν). As expected, CTMM power increases with read depth (Supp Fig 11B) and the number of cells (Supp Fig 11D).

In addition to the full description in Supp Note 3.4, we summarize these results in the Main text (lines 338-341):
 Finally, we conducted simulations at the level of single cells to evaluate the impact of sequencing depth and the number of cells (Supplementary Note Section 3.4). We found that CTMM was robust across a realistic range for sequencing depth and number of cells (Supplementary Figure S11 A, C), though power improved with greater read depth or number of cells (Supplementary Figure S11 B, D).

a) By aggregating on the pseudo bulk level the authors make use of the central limit theorem to motivate the use of normal distributions. Please quantify what is the minimal number of cells needed, so that this approximation is reliable.

We think this is asking about empirical guidance, which we agree is crucial for users. (There is no formal mathematical answer, as the CLT is about rates of asymptotic convergence in distribution.) We recommend that users require 10 cells per (individual, cell type) pair, as we do in our real data analyses. We now clearly state this important point (lines 238-239):

Requiring more than 10 cells is our default guidance.

This is also used in Crowell et al 2020 NatComm—"Subpopulations with less than 10 cells in any sample...were excluded"—though we could not find an explicit recommendation in that work.

Separately, the new simulations varying #cells and #reads (Supp Fig S11) provide helpful context for users who wish to explore this cutoff value in their own specific applications.

b) In addition, also the expression levels are quantified reliably only for the genes for highest expression levels. The other genes often have low counts and it is difficult to distinguish lowly expressed genes from non-expressed genes. what is the effect of mean expression (and number of reads per cell) on the power and accuracy of the approach?

This is a great point that overlaps R1's minor point 2. Both reviewers are exactly right: CTMM power is higher for genes with higher expression (Supp Fig 30A, copied below).

c) please also demonstrate that the tool can be applied to data produced using the 10X genomics platform, as this is one of the most widely used at the moment and differs in terms of read depth from Smartseq

We applied CTMM to the OneK1K data, which uses 10X sequencing. This overlaps with part of R1's major point 1. We copy that text below:

New analysis of real droplet-based data

We complemented our new simulations with an analysis of the OneK1K data from Yazar et al 2022 Science. This dataset is representative of next-generation scRNA-seq datasets, e.g. the sequencing is droplet-based (10X) and, depending on QC choices, N is ~500 to ~1,000 and C is ~6 to ~14. Our analysis is fully described in **Supplementary Note Section 4 ('PBMC analysis')**. The transcriptome-wide CTMM estimates and tests are shown in **Figure S22** and **Figure S20** (both copied below). We describe our results in a new Results section (lines 452-489, copied below). **Overall, we find that CTMM can be applied to recent 10X-based data.**

Application to peripheral blood mononuclear cells

The iPSC data¹³ we analyzed above used plate-based sequencing. We next sought to confirm that CTMM is applicable to droplet-based sequencing, a different scRNA-seq technology that generally trades off a greater number of cells at the cost of lower read depth. We thus applied CTMM to the recent droplet-based data from the OneK1K cohort²². This dataset has many more individuals than the iPSC data (N=982 vs 125) and more cells per individual (1,300 vs 300), but it has far fewer reads per cell (~3K vs ~0.5M). The cells themselves are also very different, as OneK1K contains peripheral blood mononuclear cells (PBMCs) from living people rather than differentiating iPSCs from a controlled lab experiment. Another important difference is that the PBMC cell types are computationally inferred, while the iPSC cell types are defined by experimental days. Finally, the PBMC cell type proportions vary substantially (Supplementary Figure S19). Our primary analysis restricted to cell types with at least 10 cells in at least 90% of individuals, resulting in 7 cell types (CD4_{NC}, CD4_{ET}, CD8_{ET}, CD8_{NC}, NK, B_{IN}, and B_{Mem}, Supplementary Note Section 4).

We find that CTMM provides powerful and robust estimates in the OneK1K data. First, CTMM detected significant cell type-specific interindividual variance for 2,310 genes out of the 11,526 total genes tested ($p < .05/11526$, Supplementary Figure S20). The top signal is RPS26 ($p = 1.78 \times 10^{-167}$, Supplementary Figure S21), which plays a key role in regulating T cells^{47,48}. Further, genetic variation causes interindividual variation in this gene that is cell type-specific²² and is linked to complex traits such as eczema and asthma⁴⁹. Second, CTMM partitions transcriptome-wide interindividual variation into components that are shared across cell types (10.9%) vs cell type-specific (21.5% on average across cell types, Supplementary Figure S22). This is an interesting contrast with the iPSC results, where shared interindividual variation was near zero. Biologically, this shows that the shared interindividual context varies far more for PBMCs than for iPSCs, which is plausible because the former are from living people while the latter are from a controlled experiment. Finally, we evaluated CTMM's Full model transcriptome-wide to quantify interindividual covariance between cell types. These estimates recapitulated expected relationships between cell types (Supplementary Figure S23). For example, the most-correlated cell types are CD4_{NC} and CD4_{ET}; intuitively, this means that an individual with above-average expression of a gene in their CD4_{NC} cells will typically also have above-average expression in their CD4_{ET} cells. The second most-correlated cell types are CD4_{NC} and CD8_{NC}, which is consistent with the observation that CD4_{NC} and CD8_{NC} shared the most genetic effects in prior work²². We next tested the robustness of CTMM to rarer cell types in a secondary analysis that includes two additional cell types, Mono_C and CD8_{S100B}. CTMM gave consistent results for the 7 larger cell types that are included in both analyses (Supplementary Figure S24-S28). As expected, CTMM's estimates are

noisier for Mono_C and CD8_{S100B}, which are rarer cell types. Nonetheless, adding these cell types enables CTMM to discover new differentially-variable genes. For example, the top newly-significant gene is TMEM176B ($p = 5.45 \times 10^{-63}$ vs $p=0.18$ in our primary analysis), which makes sense as this gene is primarily expressed in Mono_C (Supplementary Figure S29). We conclude that CTMM's results are robust to variations in the input cell types, but its estimates are less accurate for rarer cell types.

Figure S20. Transcriptome-wide distribution of CTMM p values in OneK1K using HE. CTMM's test for cell type-specific interindividual variation is on the y-axis; its test for mean differential expression across cell types is on the x-axis. Each dot represents a gene, and colors reflect the density of genes in the area, with yellow indicating higher density. Dashed lines indicate the Bonferroni significance thresholds. Top right insert: the distribution of p -values from CTMM's interindividual variance test is shown for genes where the p -value for mean differential expression is essentially 0.

Figure S22. Transcriptome-wide distribution of CTMM's estimates for cell type-specific interindividual variance in OneK1K. The homogeneous variance shared across cell types (σ_α^2) and the cell type-specific variance (V_{CT}) are estimated from the Free model using HE. Violins were truncated to $(-2, 2)$ for visibility. Cell types are ordered by cell type proportion. Interior box plots show the median and the first and third quartiles. The whiskers extend to values within 1.5 times the interquartile range from the first and third quartiles.

Minor:

4) The description of the overall pseudo bulk is quite extensive. Please motivate, why this model is useful in comparison to the much more powerful cell type specific pseudo bulk model.

In hindsight, we agree, and we appreciate this writing comment. We began our study with this data type but we realized it is so underpowered as to be almost useless (in the scenarios that we consider). Nonetheless, it has some utility because it is a semi-standard data type, it is analogous to bulk RNA-seq, and it can model continuous cell types. Therefore, we decided to keep it in the paper but to clearly acknowledge its limitations in the Introduction (lines 58-60):

We also develop a version of CTMM for overall pseudobulk data, which averages over all cells from all cell types for each individual. This is a useful analogy to bulk sequencing data, but we find that it is far less powerful in our setting.

5) In the introduction, when motivating the approach: make sure to make the distinction between sample level and cell level DE more clear.

This is a good point, as it is subtle yet essential to understanding our work. This is especially true given that it is common to study cell type differences in means—as in prior LMMs applied to scRNA-seq—but not in terms of interindividual variation—as in CTMM. (We think this is what the reviewer means by “cell level” and “sample level”.) We improved the Introduction to emphasize the concept of interindividual variation and that partitioning interindividual variation is the primary novelty of CTMM, which also addresses R1’s minor point 3.

6) How does the performance / accuracy of the model change for larger number of cell types?

This is a great question and overlaps part of R1’s Main point. We copy the shared answer below:

1) Number of cell types, C:

We added a simulation that varies C from C=4 up to C=12 (**Supplementary Figure S7**, shown below). As desired, CTMM’s primary test is calibrated for all C (panel A, ‘REML (JK)’) and its power increases with C (panel B). (While the LRT is inflated for larger C, we do not use this standard test in CTMM because it is inflated in practice, see Fig 2.) We summarize in the main text (lines 298-300):

As expected, the power increased when the main cell type became more common, when additional cell types were included, or when cell type-specific variance increased (Supplementary Figure S7).

7) The authors state that: "REML (JK), that is jackknife-based Wald test in REML, and HE were slightly inflated in CTP when sample size was 100 or lower." Sample size is likely to stay below 100 in many real life data sets. So what to do about it?

This is a great point. We have made two changes to address this. First, we now explicitly emphasize that CTMM is not appropriate for very small datasets, say $N < 50$. (We now use 50 rather than 100 because we expanded our simulations to give a more refined answer; see also our changes based on your point 8 below.)

Second, we added a supplementary analysis showing a new, simplified version of CTMM that is calibrated for N as low as ~20 (Supplementary Figure S31, copied below). This simplified model assumes that all cell types have the same level of cell type-specific variance. Although this is a valid test of heterogeneity and richer than prior models—which have not jointly fit cell type-shared and cell type-specific components—it is not our preferred

model because it is incapable of learning which cell types drive the variance differences.

Third, CTMM's estimates remain unbiased for small sample sizes (Figure S2). Only its statistical test may be miscalibrated—these tests were not even evaluated in the prior LMM applications to scRNA-seq we reference.

We have added these observation into a new sentence in the Discussion (lines 540-545):

While this inflation is small for sample sizes around ~100 and vanishes for sample sizes above ~300, CTMM is not reliable for sample sizes below ~50. Nonetheless, CTMM's estimates remain unbiased (Supplementary Figure S2), hence it can be used to profile transcriptome-wide averages for any sample size. Also, we developed a simplified version of CTMM which remains calibrated for sample sizes ~50, but it assumes that all cell type-specific variances are equal (Supplementary Figure S31).

In the discussion the conclusion is that jack knife is the most robust. How does this fit with the observation of inflation?

We agree that this was previously unclear. Our above edits now clarify that this is only for small N..

8) Results of Figure 2:

- What is the "effect size" for these results: Its true positive rate reached above 80% even when sample size was only 20 and reached 100% when sample size was 50.

To clarify, the 'effect size' is the cell type-specific interindividual variation. Note that this parameter does not exist in prior LMM studies of scRNA-seq data, which either assume it is zero (e.g., Tung 2017, Cuomo 2020, Crowell 2020) or mix it together with variation that is shared across cell types (e.g., Martinez-Jimenez 2017, Cuomo 2022). Mathematically, this is a diagonal entry in our "V," which is the covariance matrix across columns of Γ (eq 1); equivalently, this can be described as the size of cell type-specific contributions to interindividual variance. When this parameter is zero, interindividual differences do not depend on cell type. While this is mathematically simple, it is obviously biologically implausible.

If the reviewer's point is that Fig 1 uses effect sizes that are too large, we agree (if we understand this correctly, they mean Fig 1, not Fig 2). Thus, we decreased the effect sizes (i.e., v_j , the interindividual variance specific to cell type j):

Old: $v_1=2.26$, $v_2=0.56$, $v_3=0.14$, and $v_4=0.04$

New: $v_1=1.60$, $v_2=0.80$, $v_3=0.40$, and $v_4=0.20$

Now, the largest of these values (v_1) explains ~80% of variation in that cell type, which corresponds to the ~60% quantile of the real Cuomo data. The smallest (v_4) explains ~30% of variation in that cell type, which corresponds to the ~3% quantile of the Cuomo data. (We chose these quantiles to be conservative.)

Separately, Figure 2B (copied below) explicitly varies the effect size on the x-axis, providing a complementary view on the relation between effect size and power. To make this more interpretable, we added vertical lines showing the 10th and 50th percentiles of our real V estimates in the Cuomo 2020 data. We think that parameterizing by the real data quantiles helps clarify the meaning of these subtle parameters.

- What is the interpretation of the variances in figure S3 and S6? What is its scale? Would it make sense to express them relative to the overall variance?

We appreciate you catching this— σ^2_{het} was outdated notation. We have fixed this x-axis, which is now called \bar{v} (the average of v_1, v_2, v_3, v_4 after weighting by cell type proportion). To improve clarity, we have explicitly added this to the captions for Figure S3 and Figure S7 (originally Figure S6):

S3: \bar{v} can be interpreted as the proportion of variance in the overall pseudobulk due to cell type-specific interindividual variation.

S7: ...and \bar{v} , the proportion of overall variance explained by cell type-specific interindividual variation (as in Figure S3).

And yes, the reviewer is correct—because we always normalize overall pseudobulk variance to 1, this can indeed be interpreted as the proportion of overall variance.

- HE also had good power with about 50% of positive rate. Is 50% really a good power?

We agree that 'good' is subjective, so we changed the word to 'intermediate'.

- Figure 2: please make the x-axis comparable, so that one can see the trade off between true positive and false positive rate for the same measure of variation ("effect size").

In Figure 2, the x-axes of each panel should not be compared. In Fig 2a, the x-axis varies the degree of model misspecification (the noise in our estimates of ν). In Fig 2b, the x-axis varies the effect size, i.e., the amount of cell type-specific variation. In other words, Fig 2a is about the false positive rate, and Fig 2b is about the true

positive rate. (In other words, it is impossible to evaluate the false positive rate when varying the effect size, because the alternative hypothesis is true whenever the effect size is nonzero.)

- Could you also create a figure with FDR and PPV plotted in the same plot.

We have now added this plot as **Supplementary Figure S10** (note that this differs from an ROC curve, which plots true positive rate against the false positive rate varying a significance threshold). This requires simulating a mixture of true and false positive signals, and we chose to make a 50:50 mixture. The “false” signals are drawn from the Hom model, as in Fig2a (effect size $v_j = 0$ for all cell types). The “true” signals are drawn from the Free model in Fig2b with effect size $v_j = 1$ for all cell types. We then apply CTMM to all of these genes and test the Free vs Hom models (i.e., we test for cell type-specific interindividual variation). We repeat this process for 6 simulated datasets, each time using a different value for $CV(\nu)$ (as in Fig2a).

9) Is there a way to estimate how large the noise of this parameter estimate v_{ic} is in reality? This would be quite important to understand in which conditions the method really works. Is there some kind of diagnostics that one could use on a data set to see if the model would be expected to work on that data set?

We agree this is essential. CTMM treats this estimated parameter as known, which could cause false positives in theory. Fortunately, Fig 2A shows that this bias is negligible in practice. More specifically, unless the level of noise far exceeds what we ever observe in practice, there is no inflation in our REML (JK) test. (Note that

ν_{ic} is just the ordinary estimate of empirical variance across cells per (individual, cell type) pair, which helps clarify why it is so robust even when n_{ic} is as low as 10 cells.)

Please also note that Fig2A also shows why off-the-shelf LMM tools fail for CTMM's test. The standard LRT is severely biased by noisy ν . Intuitively, the LRT does not model measurement error, hence it misattributes this noise to real interindividual variance. Similar biases apply to the ordinary Wald test. Our jackknifed Wald test, however, is robust. This demonstrates why it is important to develop and carefully test novel methods when partitioning variance, which is more statistically challenging than ordinary differential expression tests.

10) Would it make sense to optimize the v_{ic} in the ML model? The single cell estimate could serve as a starting value, or it could be used to define a prior on the parameter in a Bayesian setting

We agree with the spirit, as our estimates of ν_{ic} are noisy. Unfortunately, it is impossible to optimize ν_{ic} in a maximum likelihood framework because it is unidentified (ν has the same dimension as our data, Y). The Bayesian idea is definitely reasonable and a great idea for future work, as is an empirical Bayesian variant to share information across genes. This is exactly the sort of work we hope to inspire, much as ordinary tests of differential expression inspired much of the development of empirical Bayes methodology.

11) When doing the variance decomposition there is 14% variance between individuals within celltypes, 12.3% difference in mean expression (is this also cell type specific or individual specific) and 9% noise. Does this mean that the remaining variance is due to cell type composition?

The contribution of cell type proportion due to mean expression differences across cell types is the 12.3% number. (Mathematically, this is the size of the $P \beta$ term in Supp Eq 3, which is the product of cell type proportions (P) and cell type-specific mean expression (β .) Intuitively, it is appropriate that these terms are intertwined—differences in proportion don't matter if all cell types have the same mean (ignoring the variance part), and differences in mean don't matter if all individuals have the same cell type proportions.

We now clarify that the remaining variance is due to technical covariates, such as batch and expression PCs (lines 356-360):

Weighted by cell type proportions, cell type-specific variation explained 14% of interindividual variation on average across the transcriptome (Supplementary Note Eq 3). Additionally, cell type proportion differences explained 12%, and residual cell-level variation (ν) explained 10%. The remaining variation is explained by covariates, especially PCs of pseudobulk gene expression (39%) and batch effects (21%).

Could subcelltype composition explain part of the 14%?

This is a great example of a biologically meaningful component of variation within a cell type. We thank the reviewer for this interpretation and have added it immediately after the above description of transcriptome-wide variance components. This is helpful for interpreting these complex parameters (lines 360-363):

Note that cell subtype variation within an individual will be captured in the residual variation in ν , which also captures measurement errors (i.e., RNA transcripts that exist but are not sequenced), while interindividual variation in cell subtype proportions will be captured in the interindividual covariance, V .

12) Genes with high covariance between cell types and high inter-individual variance could be viewed as those that are differentiating different persons consistently across time. Are those genes known to be responsible for different developmental outcomes or do they share any other specific functional roles?

Yes, this is exactly the correct interpretation for genes with high interindividual covariance across cell types. Mathematically, this is quantified by the “Hom” variance, i.e., σ^2_{α} . Note that prior LMMs cannot estimate this term unbiasedly because they cannot distinguish cell type-specific vs -shared.

But one of the main conclusions of our analysis of the Cuomo 2020 data is that Hom variation is negligible (“Transcriptome-wide, we found that the variation across individuals was almost entirely cell type-specific, as the homogeneous variance has median close to 0 (*median* = 0.4%, Figure 3A)”). That is, there are few genes that consistently differentiate individuals across time. This is why we only investigated the top genes from CTMM that vary across individuals in a cell type-specific way, and we do indeed find that they are enriched in biologically plausible roles (Table 2).

Reviewer #1 (Remarks to the Author):

I appreciate that the authors have performed a substantial revision of the manuscript, including extensive new analyses. I am satisfied that all of my concerns have now been addressed and I have no further comments.

Reviewer #2 (Remarks to the Author):

Chen and Dahl have revised their manuscript and added important clarifications and additional analyses. While the application to 10X data works, it seems to show that most genes that show inter-individual differences can already be identified when analysing the mean expression levels. Therefore, one of my major points still remains. While it is technically a very interesting problem, the biological significance of these variance differences is still unclear and it really should be shown in the analysis of real data sets what new insights can be discovered. I think this is very important, since it would be a motivation for researchers to pick up this method.

Re1: While it is reassuring to see known stem cell factors among the list of genes that have variance difference and no mean difference, this finding has to be connected to a biological mechanism. I would not consider this result a "new form of cell type-specific biology" without such a connection to mechanism. In their response the authors state that these genes are "enriched in plausible biological functions (Table 2)", however no testing for enrichment is actually reported in the table.

Similarly, the finding on the RPS26 gene is interesting, but this gene has already previously been identified as a strong eQTL using single cell data, with genetic variation driving the mean differences between individuals. What does the finding on increased inter individual variance add here?

What might help in this context is to systematically assess the biological functions of genes with inter individual variance differences. Are they from specific pathways? Do they share any other properties?

Specifically in the RPS26 example but also more generally in the 10X data, could it be the case that the approximations made in the model are not able to disentangle the mean - variance relation that is present in the count data? Could it be that the differences in variance that are detected are in fact consequence of the mean differences and the count nature of the data?

Re2: My point regarding the use of muscat was that this work established and evaluated a simulation framework that can already produce multi-sample single cell data that resembles the original data. The DP and the DM scenario that is proposed there can produce count distributions that have either a difference in mean between groups or a difference in variance (the bimodal mixtures can have higher variance but the same overall mean as a unimodal distribution). Since this or newer simulation frameworks have already been developed it would be good to use them to produce comparable results. Of course the authors can choose to develop their own simulation framework, but in this case they should adopt the best practices and insights from previous studies.

In particular in the new simulation: why is the binomial distribution used? Almost all single cell count models are based on the negative binomial distribution that is suitable for over-dispersed data (zero-inflated in the case of smart seq2). The authors need to show that the simulated data resembles the actual data for example using countsimQC: Sonesson, C. & Robinson, M. D. Towards unified quality verification of synthetic count data with countsimQC. *Bioinformatics* 34, 691–692 (2018).

Re3a:

Please provide a justification for this guidance. What would be the performance if all (cell type,

individual) pseudo bulks would be computed from 10 cells only?

Re3b:

It is expected to see that more genes with higher expression levels are detected to have differences in variances. The question is: do the bottom 1000 genes really have no difference in variance or are they just not detected (false negatives). This could be addressed using the simulated data.

Re3c:

It seems that the 10X data shows that almost all of the inter-individual differences can already be discovered when analysing their mean expression. What is the added value of the analysis of variance differences in this case?

Re7:

The fact that the model requires sample sizes larger than 50 means that it can currently be applied only to very few data sets. Of course this might change in the future.

Figure 4: please highlight all genes discussed in the text (including NDUFB4)

The new results show that the "shared interindividual context varies far more for PBMCs than for iPSCs, which is plausible because the former are from living people while the latter are from a controlled experiment". Please elaborate why this is plausible?

Maybe population structure or batch effects explain the higher degree of shared interindividual variance?

Chen and Dahl have revised their manuscript and added important clarifications and additional analyses. While the application to 10X data works, it seems to show that most genes that show inter-individual differences can already be identified when analysing the mean expression levels. Therefore, one of my major points still remains. While it is technically a very interesting problem, the biological significance of these variance differences is still unclear and it really should be shown in the analysis of real data sets what new insights can be discovered. I think this is very important, since it would be a motivation for researchers to pick up this method.

We are glad that our first revision addressed most of the reviewer's initial concerns, including:

- We added an entirely new real dataset (Yazar 2022, Fig S20-30) to show CTMM “works” for 10X data
- We clarified CTMM's novelty over prior LMMs (Table S3) to show it is “technically...very interesting”

However, the reviewer is not convinced CTMM has biological utility. Our three broad answers are:

1. The statement “most genes that show inter-individual differences can already be identified” compares apples to oranges: put simply, CTMM seeks *variance* differences, but prior tests seek *mean* differences.
2. Our prior resubmission does have several biologically significant results in two datasets, including:
 - a. CTMM genes are linked to pluripotency/differentiation (Tables 1, 2) and eQTLs (*RPS26*)
 - b. CTMM unbiasedly partitions interindividual variation across cell types (Figs 3, S23, S24)
3. Nonetheless, we fully agree that more biology is always better, and biology is crucial in genomics method development. To improve the biological significance of our work, our new resubmission adds:
 - a. The reviewer's requested GO enrichment tests (Tables S1, S2), which give plausible results
 - b. Enrichments in genomic and evolutionary features (Figs 4B,C, and S19). We find that constrained genes differ *less* in mean across cell types yet *more* in interindividual variance. **This clearly illustrates a biologically significant difference between mean and variance signals**

Overall, these new analyses have strengthened our demonstration that CTMM finds biologically-meaningful signals that are qualitatively distinct from mean-based tests of scRNA-seq data.

Please note that we have added some sub-numbers (in red) to ensure that we respond to each sub-point.

Re1:

1. While it is reassuring to see known stem cell factors among the list of genes that have variance difference and no mean difference, this finding has to be connected to a biological mechanism. I would not consider this result a “new form of cell type-specific biology” without such a connection to mechanism.

This language is in our prior review response, not our paper. If it were in our paper, we agree that more precise language would be appropriate, eg “quantifies a new statistical dimension of cell type-specific biology”

2. In their response the authors state that these genes are “enriched in plausible biological functions (Table 2)”, however no testing for enrichment is actually reported in the table.

This is a helpful writing comment, and we agree that we abused the word “enriched.” We revised this to:
have plausible biological functions (Table 2)

(We also added a GO enrichment test, please see Re1.4 below.)

3. Similarly, the finding on the RPS26 gene is interesting, but this gene has already previously been identified as a strong eQTL using single cell data, with genetic variation driving the mean differences between individuals. What does the finding on increased inter individual variance add here?

Differential variance vs eGenes is also apples vs oranges. The goal is not to “identify” genes, but rather to understand them, which requires many complementary analyses. In this case, CTMM unbiasedly quantifies cell type-specific interindividual variation, and the eQTL is a single SNP that contributes to this variation. Each analysis informs a distinct dimension of this gene’s biology.

4. What might help in this context is to systematically assess the biological functions of genes with inter individual variance differences. Are they from specific pathways?

To answer this question, we performed a standard GO enrichment analysis. The results are in Tables S1 and S2 and are described in the Main text (lines 245-249 and 331-334):

We next performed GO enrichment analysis using clusterProfiler [Wu et al. 2021 The Innovation]. We tested the top 100 CTMM genes that did not have significant differences in mean ($p > 0.05$ after Bonferroni correction). We found dozens of significant enrichments, almost all of which reflect cellular metabolic activity (Table S1). This finding aligns with the known importance of variation in metabolic state during iPSC differentiation [Xu et al. 2013 Cell Metab].

As in the iPSCs, we tested GO enrichment in the top CTMM genes. In this large dataset, almost all genes have significant differential mean expression, so we tested the top 100 CTMM genes irrespective of their mean differences. Almost all of the top enrichments relate to immune function, including several that are specific to leukocytes (Table S2).

5. Do they share any other properties?

Please recall that our initial revision described several properties enriched in top CTMM genes, including properties requested by R1 (comment #2) and by R2 (comment #3b). We showed these results in Fig S31.

Nonetheless, to further strengthen our paper, we tested gene features related to enhancers and selection, as suggested by our (biologist) colleague. Broadly, we find many enrichments of CTMM parameters in these features (Fig S19). **As one highlight, CTMM results have an opposite association with selective constraint than ordinary differential mean expression (Fig 4C). This is a concrete insight that clearly cannot be gained solely from standard analyses of mean expression.** Below, we copy our new additions to the Results (lines 295-311), Methods (lines 623-644), and figures:

Gene features associated with cell type-specific variation [Results]

We next evaluated the relationship between CTMM results and 4 gene features related to genome structure and evolution. We compared CTMM’s measure of cell type-specificity, which is based on interindividual variance, to a standard measure of cell type-specificity based on mean differences (Methods). First, we found that both CTMM and ordinary differential expression signals were enriched in genes with larger enhancer domains (based on the number of enhancers or enhancer domain score [EDS], Figure 4B and Supplementary Figure S19). These results align with previous findings that genes with larger enhancer domains were less likely to exhibit ubiquitous expression across tissues [Wang & Goldstein 2020 AJHG]. Second, we examined a measure of gene conservation called LOEUF

(loss-of-function observed/expected upper bound fraction). We found that more-constrained genes have lower mean differences across cell types ($p = 1.0e-3$, Figure 4C), consistent with previous findings that constrained genes were more frequently ubiquitously expressed across tissues [Lek et al. 2016 Nature, Karczewski et al. 2020 Nature]. CTMM's cell type-specificity measure also correlates with LOEUF, but in the opposite direction ($p = 1.2e-8$, Figure 4C). Digging deeper, CTMM shows this primarily results from decreases in cell type-shared variation (Supplementary Figure S19). In other words, stronger selection implies that cell types and individuals are more constrained toward their averages, so cell type-specific interindividual variation plays a larger role. We found qualitatively similar results using pLI (Supplementary Figure S19).

Enrichment of gene features related to enhancers and selection [Methods]

We evaluated four gene-level features: LOEUF, pLI, EDS, and the number of enhancers. LOEUF and pLI were obtained from the Genome Aggregation Database (gnomAD) version 2.1 [Karczewski et al. 2020 Nature]. LOEUF and pLI measure a gene's susceptibility to loss-of-function mutations, and they approximately quantify the degree of selection on a gene. EDS and the number of enhancers were obtained from Wang and Goldstein [2020 AJHG]. The number of enhancers was computed from enhancer-gene links inferred by Liu et al. [2017 Genome Biol] based on chromatin state and correlation of histone modifications with gene expression. EDS is a comprehensive score derived from 108 features associated with enhancer domains, including the number of enhancers. It reflects the size and redundancy of enhancer domains in a gene.

For each feature, genes were stratified into deciles based on their respective feature scores.

Subsequently, we computed both the mean and median values of various gene expression properties within each decile, as well as their standard errors. These gene expression properties are:

- the total interindividual variance, which sums the cell type-shared variance with the average cell type-specific variance: $\sigma_{\alpha}^2 + \tilde{V}$, where $\tilde{V} = \frac{1}{C} \sum_{c=1}^C V_{cc}$ is the average cell type-specific variance.
- the proportion of interindividual variation that is cell type-specific, defined by $\frac{\tilde{V}}{\sigma_{\alpha}^2 + \tilde{V}}$.
- the amount of mean differences across cell types, quantified by the variance of the mean expression level across cell types: $var(\beta)$.
- the positive rate for two cell type-specificity tests: CTMM's test of cell type-specific interindividual variance and the ordinary test of cell type-specific mean expression.

To robustly examine the broad relationship between CTMM results and gene features, we performed meta-regression of each decile's mean and median CTMM results against the decile index using ordinary least squares.

Figure 4. Contrasting CTMM with ordinary differential expression transcriptome-wide. (A) Each point conveys the cell type-specificity of a gene, where the x-axis tests for cell type-specific means (ordinary) while the y-axis tests for cell type-specific interindividual variance (CTMM). Genes described in the main text are highlighted, and Bonferroni-significance is illustrated as dashed horizontal and vertical lines. All results are from CTMM fit using REML and tested using jackknife. (B, C) Plots quantify cell type-specificity of interindividual variance (blue, left) and mean expression (orange, right) compared to each decile in the transcriptome of EDS (B) or LOEUF (C). Each point is the average across all genes in that decile, and error bars display one standard error. p-values correspond to the meta-regression of these mean points. The x-axis ticks show the median value of each feature in each decile.

Figure S19. Gene features associated with CTMM's interindividual variance test and ordinary mean differential expression across cell types. Each column shows one of four gene features divided into deciles, with x-axis ticks showing the feature's median value within each decile. LOEUF and pLI measure a gene's susceptibility to loss-of-function mutations, where lower LOEUF and higher pLI correspond to higher intolerance. Dashed lines in pLI mark pLI = 0.9, a common cut-off for highly constrained genes (pLI > 0.9). Enhancer number and EDS measure the enhancer structure nearby a gene, where larger enhancer numbers and larger EDS indicate larger enhancer domains. Rows show five gene expression properties from CTMM corresponding to interindividual variance (rows 1-3) and mean expression (rows 4-5) across cell types in the

iPSC data. (1) The total interindividual variance combines cell type-shared and -specific variance: $\sigma_{\alpha}^2 + \tilde{V}$, where $\tilde{V} = \frac{1}{c} \sum_{c=1}^c V_{cc}$ is the average cell type-specific variance. (2) The proportion of interindividual variance that is cell type-specific, defined by $\frac{\tilde{V}}{\sigma_{\alpha}^2 + \tilde{V}}$. (3) The proportion of genes with significant cell type-specific interindividual variance. (4) The variance of cell type-specific mean expression ($var(\beta)$). (5) The proportion of genes with significant mean differences across cell types. Displayed regression p values correspond to a simple meta-regression of each decile's mean or median against the decile index. Solid lines indicate significant meta-regressions at $p < 0.05$, while dashed lines indicate $p > 0.05$.

6. Specifically in the RPS26 example but also more generally in the 10X data, could it be the case that the approximations made in the model are not able to disentangle the mean - variance relation that is present in the count data? Could it be that the differences in variance that are detected are in fact consequence of the mean differences and the count nature of the data?

Generally, we agree that mean-variance dependence is crucial in count data. But it is important to distinguish components of variance. Mean differences across cell types do cause variance differences across cell types—but this effect is identical across individuals, hence it does not inflate interindividual variance. We added a new simulation to show this (Fig S32A, below). (Note that this concern does apply to *muscat*, which does not distinguish these components of variance; please see the “Details” in our response to Re2.1 below.) Overall, this concern is very important for intraindividual variance, but not for interindividual variance.

Nonetheless, there is a case where CTMM has slight inflation. We have always noted this in our Discussion (copied below), where we also explain why this is unlikely to be important in practice. Specifically, mean differences cause slight inflation under (1) cell type-shared interindividual variance ($\sigma_{\alpha}^2 > 0$) *and* (2) low expression levels *and* (3) large mean differences (Fig S32B, below). For example, 2-fold differences do not cause inflation, but with 4-fold differences the observed FPR is 7.8% at a nominal 5% FPR. Importantly, this inflation is limited to low-expressed genes, which we do not see in practice (as Re3b points out below).

We conclude that our initial Discussion caveat is appropriate: Mean-variance dependence can cause slight inflation, but this is not important in practice. Thus, we appended these new simulations to our prior Discussion statement (new content in blue, lines 399-400). We appreciate the reviewer bringing this up and agree this is an important addition—our new simulations bolster and quantify our previously-vague statement:

Fourth, it is well-known that count data evince a complex mean-variance relationship, and studies have observed that the variance of gene expression across cells is dependent on mean expression [Eling et al]. Nonetheless, simulations show that this problem is unlikely to be important in practice (Supplementary Figure S32). Moreover, we find biologically plausible genes with significant differential variance but without significant differential mean, showing that modelling variance has utility beyond merely tagging mean signals.

Figure S32. False positive rate in CTMM due to mean-variance dependence across cell types.

Simulations use the same framework as in Figure S11 with known v (left side of Figure S11, panels A and C). (A) CTMM is unbiased regardless of the fold change in the absence of true interindividual variation (Null model). To realistically simulate from this Null model, we shuffled cells across individuals to remove all interindividual variation. (B) CTMM can be inflated for low-expressed genes under the Hom model, where interindividual variation does exist but is entirely shared across cell types. Low/Medium/High indicate tertiles of total expression across the studied genes.

Re2:

1. My point regarding the use of muscat was that this work established and evaluated a simulation framework that can already produce multi-sample single cell data that resembles the original data. The DP and the DM scenario that is proposed there can produce count distributions that have either a difference in mean between groups or a difference in variance (the bimodal mixtures can have higher variance but the same overall mean as a unimodal distribution). Since this or newer simulation frameworks have already been developed it would be good to use them to produce comparable results. Of course the authors can choose to develop their own simulation framework, but in this case they should adopt the best practices and insights from previous studies.

This is incorrect. *muscat's* DP and DM models cannot simulate cell type-specific interindividual variance without mean differences. We described this in our initial review response and elaborate further below.

Also, *muscat* cannot address other comments from the reviewer, e.g., read depth is not an input parameter.

(Details: The variance in *muscat* is cell-to-cell variance (our ν), or intraindividual variance. This is orthogonal to CTMM's focus, which is interindividual variance (our σ_{α}^2 and V). Mathematically, individuals could be identical on average across cells ($\sigma_{\alpha}^2=V=0$) yet have huge variance across cells ($\nu \gg 0$); conversely, individuals could be very different ($\sigma_{\alpha}^2, V \gg 0$) yet have no variation amongst their cells ($\nu=0$). Partitioning these distinct components of variation in scRNA-seq data is the goal of CTMM.)

2. In particular in the new simulation: why is the binomial distribution used? Almost all single cell count models are based on the negative binomial distribution that is suitable for over-dispersed data (zero-inflated in the case of smart seq2).

This is a misunderstanding. We do draw from a binomial distribution conditional on the number of reads in a given cell, but marginalizing over cells induces over-dispersion. Indeed, one way to construct the negative binomial is by marginalizing over a Poisson distribution's mixing parameter.

3. The authors need to show that the simulated data resembles the actual data for example using countsimQC: Sonesson, C. & Robinson, M. D. Towards unified quality verification of synthetic count data with countsimQC. Bioinformatics 34, 691–692 (2018).

We used this plotting package and confirmed that our simulated data closely resembles the real data (Figure S10, copied below). We have added this in the Main text (lines 185-188):

To assess whether our simulated count distribution is realistic, we compared it to the real data using countsimQC [Sonesson and Robinson]. The comparison demonstrated a good fit to the real data in terms of the mean-variance distribution and the fraction of zeros per gene (Supplementary Figure S10).

Figure S10. Comparison of count distribution between real and simulated scRNA-seq data. The real data are genes from the iPSC data, and each simulated gene is based on parameters from one of these real genes (Supplementary Note Section 3.4). We simulated 7202 cells, each corresponding to a real cell from the cell type “day0”. Each dot in the top panels shows a gene’s expression mean and variance across cells. The bottom panels show the distribution of the fraction of zeros per gene across the simulated genes.

Re3a:

1. Please provide a justification for this guidance.

It seems that our prior review response’s justifications were missed, so we copy them below:

We think this is asking about empirical guidance, which we agree is crucial for users. (There is no formal mathematical answer, as the CLT is about rates of asymptotic convergence in distribution.) We recommend that users require 10 cells per (individual, cell type) pair, as we do in our real data analyses. We now clearly state this important point (lines 238-239):

Requiring more than 10 cells is our default guidance.

This is also used in Crowell et al 2020 NatComm–“Subpopulations with less than 10 cells in any sample...were excluded”--though we could not find an explicit recommendation in that work.

Separately, the new simulations varying #cells and #reads (Supp Fig S11) provide helpful context for users who wish to explore this cutoff value in their own specific applications.

To summarize our justifications:

1. Mathematical: Not possible (the question is ill-posed)
2. Practical: It works well in two real datasets
3. Literature: We followed the *muscat* paper (Crowell 2020 NatComm)

2. What would be the performance if all (cell type, individual) pseudo bulks would be computed from 10 cells only?

This is (1) unrealistic and (2) not our guidance. Nonetheless, we (3) added a new robustness test.

(1) Real data have far more cells than the reviewer describes, by an order of magnitude (averages: 72 in the iPSCs, 92 in the PBMCs). We illustrate this in a histogram of #cells per (individual, cell type) in the iPSCs below (note that we had to cut the x-axis at 200 so the range around 10 is visible.)

(2) We do not advise studies to use 10 cells per (cell type, individual). We advise the opposite—exclude + impute such data. In the reviewer’s (unrealistic) hypothetical scenario, our advice would be to not use CTMM.

(3) Nonetheless, we agree that robustness tests add context. Thus, we asked: What if we modify the cutoff? We repeated our iPSC analysis changing the cutoff from 10 to 5 or 20. We find this has negligible impact, as the correlation in the $-\log_{10}(p)$ -values is $\sim 97\%$. We have added this to the Main text (lines 612-613):

Requiring more than 10 cells is our default guidance (in practice, we find that our results are robust to modifying this cutoff from 10 cells to 5 or 20, Supplementary Figure S35).

Figure S35: CTMM’s robustness to the threshold on the minimum number of cells per (individual, cell type) pair. When this number is not above the cutoff, the corresponding entry of the pseudobulk matrix is set to missing and then imputed. Plots compare CTMM’s p-value in our primary analysis (x-axis), which requires 10 cells per (individual, cell type), to the CTMM p-value obtained when this is decreased to 5 (y-axis, panel A) or

increased to 20 (y-axis, panel B). The numbers indicate the number of genes in each quadrant, which are defined by vertical and horizontal lines indicating Bonferroni-significance.

Re3b:

It is expected to see that more genes with higher expression levels are detected to have differences in variances. The question is: do the bottom 1000 genes really have no difference in variance or are they just not detected (false negatives). This could be addressed using the simulated data.

After carefully reading several times, we think this is a rhetorical question. Specifically, sentence 1 (higher expression=higher power) is equivalent to sentence 2 (lower expression=lower power).

Nonetheless, we agree this is important. Indeed, we previously added Fig S11 to show this point, which was requested by both reviewers, and we explicitly describe this limitation in our Results and Discussion:

power improved with greater read depth or number of cells (Supplementary Figure S11 B, D).

We find that higher overall levels of a gene's expression increase the power of CTMM (Supplementary Figure S31)

Re3c:

It seems that the 10X data shows that almost all of the inter-individual differences can already be discovered when analysing their mean expression. What is the added value of the analysis of variance differences in this case?

This is apples vs oranges. Please see our other responses pointing out why we believe that CTMM finds meaningful biology that cannot be found by analyzing means, which is most clear in our new Fig 4C.

Re7:

The fact that the model requires sample sizes larger than 50 means that it can currently be applied only to very few data sets. Of course this might change in the future.

Broadly, we agree that CTMM's full functionality requires ~50 samples, and its utility will grow with time.

Narrowly, this is incorrect, as explained by 2/3 of our prior responses, which may have been missed:

Second, we added a supplementary analysis showing a new, simplified version of CTMM that is calibrated for N as low as ~20 (Supplementary Figure S31, copied below). This simplified model assumes that all cell types have the same level of cell type-specific variance. Although this is a valid test of heterogeneity and richer than prior models—which have not jointly fit cell type-shared and cell type-specific components—it is not our preferred model because it is incapable of learning which cell types drive the variance differences.

Third, CTMM's estimates remain unbiased for small sample sizes (Figure S2). Only its statistical test may be miscalibrated—these tests were not even evaluated in the prior LMM applications to scRNA-seq we reference.

Figure 4: please highlight all genes discussed in the text (including NDUFB4)

Thank you, we made these changes and we agree they improve our Figure.

The new results show that the “shared interindividual context varies far more for PBMCs than for iPSCs, which is plausible because the former are from living people while the latter are from a controlled experiment”. Please elaborate why this is plausible?

This is helpful. We tried to put too much information into one sentence, and the rationale was not clear. We now write this out more precisely (lines 336-342):

This is an interesting contrast with the iPSC results, where shared interindividual variation was near zero. Biologically, this could be explained by differences in cellular environment: individual-level covariates like age, smoking, or BMI may have shared effects across cell types and they are likely to have larger effects on PBMCs in whole blood than iPSCs in a controlled lab.

Note that this is yet another analysis that clearly demonstrates CTMM’s novelty (and has nothing to do with “discovering” genes).

Maybe population structure or batch effects explain the higher degree of shared interindividual variance?

These are unlikely. Population structure: CTMM does not use genotype data. Batch effects: (1) we adjust for batch effects when we fit CTMM; (2) both datasets model batch effects in the same way, so this is unlikely to explain the dramatic differences we observe between datasets

Reviewer #2 (Remarks to the Author):

The authors have addressed all of the comments. In particular, I would like to thank the authors for added new analyses that demonstrate that the variance differences identified with their method are associated with different genomic and evolutionary properties.